# Emergent Communication of Generalizations

**Jesse Mu**
Stanford University
muj@stanford.edu

**Noah Goodman**
Stanford University
ngoodman@stanford.edu

## Abstract

To build agents that can collaborate effectively with others, recent research has trained artificial agents to communicate with each other in Lewis-style referential games. However, this often leads to successful but uninterpretable communication. We argue that this is due to the game objective: communicating about a single object in a shared visual context is prone to overfitting and does not encourage language useful beyond concrete reference. In contrast, human language conveys a rich variety of abstract ideas. To promote such skills, we propose games that require communicating generalizations over *sets* of objects representing abstract visual concepts, optionally with separate contexts for each agent. We find that these games greatly improve systematicity and interpretability of the learned languages, according to several metrics in the literature. Finally, we propose a method for identifying logical operations embedded in the emergent languages by learning an approximate compositional reconstruction of the language.

## 1   Introduction

The communication systems that emerge when two agents are trained to cooperate offer a window on the evolution of human language, as well as a promising avenue for improving the collaboration abilities of artificial agents. Much recent work studies Lewis-style [24] signaling games (Figure 1a), where agents are trained to refer to a single object in a shared visual context. However, a general consensus of this work is that without careful environmental pressures, agents develop successful but uninterpretable communication schemes distinctly unlike human language [1, 5, 6, 16, 20].

We argue that the reference games typically used in these studies are ill-suited to drive linguistic systematicity for two reasons. One is perceptual: agents can exploit inscrutable patterns in single inputs, which leads to communication via spurious features [3]. The other reason is cognitive: human language can convey abstract ideas, such as kinds and causes, not only reference to specific objects. Simple reference games are unlikely to drive emergence of such abstract language. In particular, *generalizations* over categories are a crucial part of language [36], helping us transfer knowledge that may be useful in the future. For example, we would like to teach our kin not just to avoid one specific lion, but to avoid all lions, including those that have not yet been seen. Some have even argued that language emerged precisely from this need to teach hard-won generalizations to others [19]. With this idea in mind, can we design an experimental setting that better catalyzes these abilities?

In this paper, we propose extensions of Lewis-style signaling games to *sets*. In the *set reference* (setref) game, a teacher must communicate to a student not just a single object, but rather a group of objects belonging to a concept (Figure 1b). In the *concept* game, each agent sees different examples of the concept (Figure 1c). Inspired by human teaching [9], our core insight is that requiring generalization to combinatorially large sets of (possibly unseen) objects encourages agents to learn and communicate rich abstractions across inputs (e.g. *seagulls*), instead of low-level features (e.g. *color #FDA448*). These tasks are more difficult than traditional reference games, and we will show with a variety of metrics that the learned languages are more systematic, compositional, and interpretable. Finally, the rich compositional space of concepts explored in these games allows us to probe for specific logical

35th Conference on Neural Information Processing Systems (NeurIPS 2021).

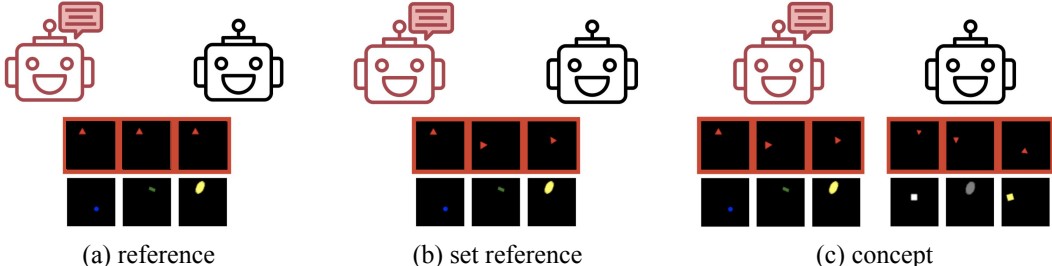



(a) reference        (b) set reference        (c) concept



Figure 1: Communication games for the concept *red triangle*. Given a set of targets (red borders) and distractors, a **teacher** must send a message to help a **student** identify the targets. In (a) reference games, targets are identical; in (b) set reference (setref) games, there are multiple targets; and in (c) concept games, the agents see different inputs.

operators in the emergent language. We propose a method for doing so, thereby demonstrating how the emergent languages reflect the compositional structure of their inputs.

## 2 Communication Games

First imagine a generic communication game between a teacher $T$ and student $S$. Let $G = (c, X^T, Y^T, X^S, Y^S)$ be a communication game, where $c : \mathcal{X} \mapsto \{0, 1\}$ is a latent concept to be communicated, $X^T = \{x_1^T, \ldots, x_n^T\}$ is a set of $n$ inputs presented to the teacher, and $Y^T = \{y_1^T, \ldots, y_n^T\}$ is a set of labels for the teachers' inputs, defined as $y_i^T = c(x_i^T)$. We call $x_i^T$ a *target* if $y_i^T = 1$, which indicates that $x_i^T$ is a member of the concept $c$; otherwise $x_i^T$ is a *distractor* and $y_i^T = 0$. $X^S$ and $Y^S$ are defined similarly for the student. Given its targets and distractors (but not the latent concept $c$), the teacher must send a message $m$ to a student that allows them to correctly identify their own targets, where $m = (m_1, \ldots, m_n)$ is a discrete sequence over a fixed alphabet $m_i \in \mathcal{M}$. Now we can define variants of this communication game as follows:

**Reference game.** In basic reference games, the teacher and student see the same examples ($X^T = X^S$, $Y^T = Y^S$) and there is a single (repeated) target: $x_i^T = x_j^T$ for all $i, j$ where $y_i^T = y_j^T = 1$.[1]

**Set reference (setref) game.** Now we extend our game to *sets*: the teacher and student see the same examples, but there are multiple target images encoding the concept (e.g. different *red triangles*).

**Concept game.** Finally, we propose the more abstract concept game, where the teacher and student see *different examples* ($X^T \neq X^S$, $Y^T \neq Y^S$) of the same concept. When $X^T$ and $Y^T$ contain a single positive and negative example, this is a reference game with separate inputs for each agent, a setup which has been shown to encourage linguistic systematicity in some settings [8, 21, 22].

## 3 Models

Now we will formalize our models of the teacher and student. Given a communication game $G$, a teacher is defined as a distribution over messages given inputs $p^T(m \mid X^T, Y^T)$, and a student is a distribution over targets given a message: $p^S(Y^S \mid X^S, m) = \prod_i p^S(y_i^S \mid x_i^S, m)$.

**Teacher.** The teacher encodes all inputs with a convolutional neural network (CNN) $f_\theta^T$; embeddings for targets and distractors are averaged to form target and distractor *prototype* embeddings [33],[2] which then conditions a recurrent neural network (RNN) used to produce the message. Let $X_+^T$ and $X_-^T$ denote the sets of targets and distractors in $X^T$; then define a prototype embedding

---

[1]For the most consistent comparison across games, our reference game has multiple identical targets and student target decisions made independently for each input, instead of the single-target forced-choice setting. Appendix E shows results with traditional games trained with cross entropy loss; conclusions are the same.

[2]Note that the teacher's job is one of representation learning for *sets* [42] and thus one can use any set representation learning method beyond the prototypical networks explored here. As a more advanced implementation, we tried a variant of Set Transformers [23], but this did not give any tangible performance benefit.

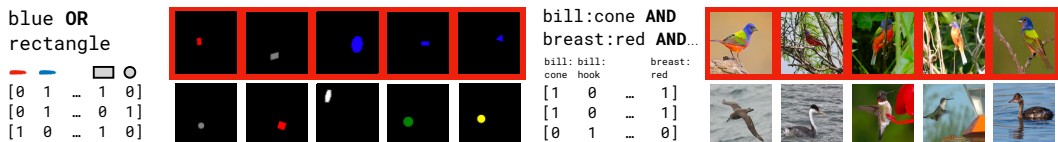

Figure 2: Example games, with targets (red border) and distractors for the ShapeWorld concept *blue OR rectangle* (left) and the Birds concept *painted bunting* (right). Concepts are represented as intensional logical formulas (top) or extensional sets of boolean input features (bottom), where each vector is the binary representation of an individual member of the concept (e.g. *blue rectangle*, or the labeled attributes of one particular bird). See Figure S1 in Appendix A for additional game examples.

$\mathbf{x}_+^T = \frac{1}{|X_+^T|} \sum_{x_i \in X_+^T} f_\theta^T(x_i)$ (analogously for $\mathbf{x}_-^T$). Then $p^T(m \mid X^T, Y^T) = p_{\text{RNN-DECODE}}(m \mid \text{proj}([\mathbf{x}_+^T; \mathbf{x}_-^T]))$ where proj is a linear projection to the RNN hidden state.

**Student.**  The student takes a message and makes predictions about the labels $\hat{y}_i^S$ independently for each input $x_i^S$. Given the teacher message and an input image, define $p^S(y_i^S \mid x_i^S, m) = \sigma(\text{RNN-ENCODE}(m) \cdot f_\phi^S(x_i^S))$, where $f_\phi^S$ is a separate CNN for the student.

We jointly train a teacher-student pair, including the vision modules and communication protocol, via stochastic gradient descent to maximize the student likelihood of selecting the correct target images. Formally, the loss for a single game is

$$\mathcal{L}(T, S, G) = -\sum_i \log p^S(y_i^S \mid x_i^S, \hat{m}), \quad \hat{m} \sim p^T(m \mid X^T, Y^T). \tag{1}$$

To maintain backwards differentiability, we use the straight-through Gumbel-Softmax [14] trick with $\tau = 1$, simulating samples from the teacher distribution over tokens via softmax and discretizing in the forward pass only. For full model and training details and a link to code, see Appendix A.

## 4   Tasks

We examine the languages developed for our proposed communication games over two datasets: first, an artificial shape dataset which allows us to evaluate communication over cleanly defined logical concepts; second, a dataset of birds to test agents' ability to learn concepts from realistic visual input.

**ShapeWorld.**  We use the ShapeWorld visual reasoning dataset [18] (Figure 2, left). For reference games, target images are a single object, defined by a conjunction of a shape and a color (e.g. *red triangle*, *green square*); of the 30 possible shapes, we hold out 20% for testing. For setref and concept games, concepts include the conjunctions tested in reference games, but also primitive concepts (e.g. *blue shapes*) and arbitrary disjunctions or conjunctions of (possibly negated) shapes and/or colors. This produces 312 concepts, 20% of which are held out for testing. These more general concepts cannot be tested in reference games, since a single object is always identified by a shape and color. This rules out disjunctive concepts like *red OR blue* that only make sense across multiple objects. Similarly, since reference game targets must necessarily have both color *and* shape, we can never guarantee that a message for a reference game only carries the semantics *blue* and not, for example, *blue circle*, if the target is a blue circle. By looking at sets, setref and concept games allow us to more precisely control the semantics of the concepts in each game.

Each game consists of 10 targets depicting shapes satisfying the concept, and 10 distractors. We specifically sample "hard" targets and distractors to test understanding of conjunctions or disjunctions (see Appendix B for details). Finally, we specify an agent vocabulary of 14 tokens and maximum length 5, so that the communication channel has the same bandwidth as the true concept formulas;[3] see Appendix C for experiments varying these parameters for both this dataset and the next one.

**Birds.**  We next use the Caltech-UCSD Birds dataset [40] which contains 200 classes of birds with 40–60 images (Figure 2, right). As before, reference games involve a single target; setref and concept

---

[3]In the true concept formulas there are 5 shapes, 6 colors, and the 3 AND/OR/NOT operators, i.e. 14 tokens; and the longest concept formulas have the form *NOT x AND NOT y*, i.e. length 5.

game targets are members of a specific bird class. We use 100 classes at train and 50 at test, sampling 5 targets and 5 distractors per game. The dataset contains boolean attributes (e.g. *beak*, *size*) for individual birds and classes.[4] Thus, we represent reference game concepts as the feature vector of the target bird, and setref/concept game concepts as the feature vector of the class. In our evaluation, we will measure how well the languages capture these features. As there is no reference language for this task, we set the vocabulary size to 20 and the message length to 8 (though again see Appendix C).

## 5    Evaluation

We first measure communication success, as defined by student accuracy on held-out games from seen and unseen concepts, with the unseen concepts testing a language's ability to generalize compositionally. Ultimately, however, we are interested in the systematicity of the learned languages, which we evaluate via the following measures:

**Information theoretic measures.**    We first ignore the specific content of messages and concepts, and simply compute simple information theoretic quantities, by treating each distinct message and concept as a unique value and imagining probability distributions over these values. First, we measure the **conditional entropy** of teacher messages given concepts, $H(M \mid C)$, averaged across seen and unseen games; lower entropy indicates that agents use more consistent language for a fixed concept. However, $H(M \mid C)$ measures systematicity only in one direction; as a more symmetric measure, we also use the **adjusted mutual information**

$$\text{AMI}(M, C) = (I(M, C) - \mathbb{E}(I(M, C))) / (\max(H(M), H(C)) - \mathbb{E}(I(M, C))). \quad (2)$$

Here, $\mathbb{E}(I(M, C))$ is evaluated with respect to a hypergeometric model of randomness, where $M$ and $C$ are assumed to be random permutations subject to the number of unique values in either set [39]. AMI $\in [0, 1]$ represents the mutual information between $M$ and $C$, adjusted for chance to maintain comparability across different distributions of messages and concepts. A higher score indicates overall higher alignment between messages and concepts.

**Topographic $\rho$.**    To more precisely measure the lexical compositionality of a language, a measure often used in the literature is *topographic $\rho$* [4, 20, 22, 25], which quantifies the agreement between two representational systems a la Representational Similarity Analysis [17]. We define a distance metric between game concepts $d_C(c_i, c_j)$ and another between agent messages $d_M(m_i, m_j)$, compute pairwise distances between concepts and between messages, and measure their alignment with Spearman's $\rho$. A high $\rho$ indicates that a teacher sends lexically similar messages for similar concepts.

For our distance function on messages, we use the **Edit** (i.e. Levenshtein) distance with equal insert/delete/replace costs. For distances between game concepts, we define two distances based on intensional and extensional representations of concepts (Figure 2). First, we use the word-level **Edit** distance between string representations of logical formulas. Second, we use the **Hausdorff** distance $d_H$, a distance function between sets of members of a concept. Let $Z^a = \{z_1^a, \ldots, z_n^a\}$ be the set of feature-based boolean representations of inputs belonging to concept $a$. For ShapeWorld, these are two-hot vectors denoting the color and shape of all objects belonging to a specific formula; for example, for the concept *red*, we have vectors for *red triangle*, *red circle*, and so on. For Birds, these are boolean vectors of the attributes of each individual bird of a species. Then the Hausdorff distance $d_H$ is the maximum distance from any point in one set to the closest point in the other: $d_H(Z^a, Z^b) = \max(\sup_i d(z_i^a, Z^b), \sup_j d(z_j^b, Z^a))$, where $d(a, B) = \inf_{b \in B} \text{EditDistance}(a, b)$.

## 6    Results

Table 1 shows test accuracy, as measured by student classification accuracy (partial credit given), on communication games over seen and unseen concepts for 5 models trained in each condition. Reference game performance is high across both datasets, and agents are able to generalize well to unseen games. Accuracy on setref and concept games is lower, with lower performance on novel games in both datasets.[5] Overall, communicating sets is a much harder task than specific reference.

---

[4]Feature vectors for individual birds in a class vary due to the visibility of features in the image; class vectors are averaged across all individual birds, then rounded to 1 or 0.

[5]To calibrate setref and concept performance, Appendix D tests listeners on ideal (human) languages.

Table 1: Student accuracy (seen and unseen concepts, where chance accuracy is 50%), conditional entropy of messages given concepts (lower is better), and adjusted mutual information score (higher is better), with (SD) across 5 runs.

| Dataset | Game | Acc (Seen) | Acc (Unseen) | $H(M \mid C)$ | $\mathrm{AMI}(M, C)$ |
|---|---|---|---|---|---|
| ShapeWorld | Ref | **97** (0.4) | **98** (0.3) | 7.3 (0.2) | 0.04 (0.00) |
| | Setref | 92 (2.2) | 87 (1.6) | 3.9 (0.6) | 0.59 (0.08) |
| | Concept | 88 (3.4) | 75 (3.0) | **2.4** (0.2) | **0.66** (0.07) |
| Birds | Ref | **93** (0.3) | **89** (0.1) | 5.9 (0.2) | 0.05 (0.00) |
| | Setref | 89 (0.2) | 78 (0.2) | 5.2 (0.1) | 0.17 (0.02) |
| | Concept | 88 (0.1) | 73 (0.3) | **4.1** (0.2) | **0.26** (0.02) |

The ability to communicate accurately, even on unseen concepts, is not necessarily indicative of more *systematic* communication; generalization without compositional language is a common finding in the literature [1, 6, 16, 22]. Instead, we find that the more difficult games produce more systematic language. For ShapeWorld, concept game entropy over messages is lower than reference game entropy (2.4 vs. 7.3), and AMI is higher (0.66 vs. 0.04), with setref in the middle; this pattern also occurs in Birds. Furthermore, Figure 3 shows that topographic $\rho$ between the languages and the (Edit and Hausdorff) concept distances is higher for concept and setref than ref, throughout training.

Figure 4 (more examples in Appendix F) shows messages generated by agents for concepts in both games, where we arbitrarily assign letters to agent "words" to assist with interpretation. For example, for concept game teachers conveying the concept *red AND triangle*, the innermost circle is largely red, indicating that the majority of messages sent for this concept begin with d; proceeding outward, we see blue regions indicating the token e. Thus, concept agents consistently use the 5-token sequence `deeee` to refer to red triangles (less commonly, `edeee`). In contrast, for different red triangles, the language of reference game agents is strikingly inconsistent, as indicated by the extreme diversity of colors, while setref language is somewhere in the middle. The entropies over the message distributions for these concepts correlate with the "busyness" of the plots, reinforcing the idea that the setref and concept languages are more consistent.

## 6.1 Set Size

The difference between reference and setref/concept games can be interpreted as a continuum, ranging from referring to a single object (reference) to referring to a potentially infinite number of objects. With a small number of objects, it may still be possible to communicate only low-level, non-generalizable features of the set, similar to the strategies adopted by our reference game agents. In contrast, increasingly large numbers of objects should put further pressures on the semantics of the messages to not refer to individual inputs, but rather entire categories.

In Figure 5, we confirm this hypothesis, showing how increasing the number of targets $n$ (with equal numbers of distractors) increases language systematicity. $n$ has a statistically significant effect on topographic $\rho$ for ShapeWorld setref (Spearman $\rho = 0.39, p = 0.029$) and concept ($\rho = 0.75, p < 10^{-5}$) and Birds setref ($\rho = 0.90, p < 10^{-7}$) and concept ($\rho = 0.51, p = 0.024$). When $n = 1$, the setref game is equivalent to a reference game with 1 target and 1 distractor, and the concept game is similar, but with agents given separate inputs. Our results suggest that this decoupling, as proposed by Lazaridou et al. [21] and often used in the emergent communication literature, promotes systematicity in some cases (Birds) *but not others* (ShapeWorld). We additionally show that (1) sets are an *alternative* way of encouraging systematicity without needing this separation, and (2) even with this separation, larger set sizes further improve the systematicity of the resulting language.

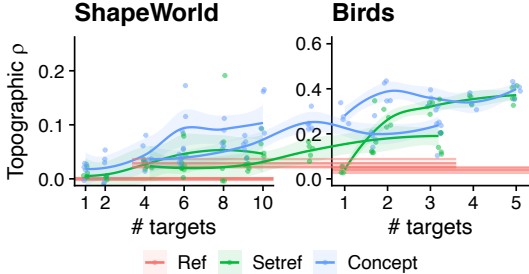

Figure 5: Topographic $\rho$ (concept Edit distance) with varying number of targets (average over seen/unseen splits). Each point and Ref line is an independent run.

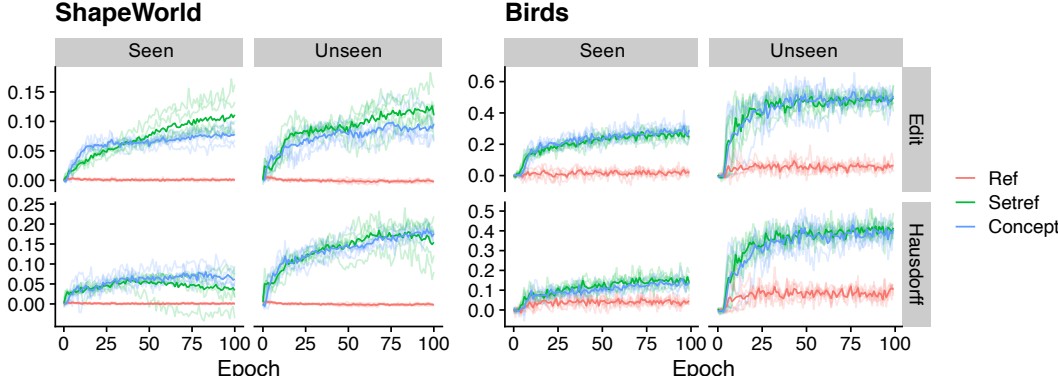

Figure 3: Topographic $\rho$ between language and Edit (top) or Hausdorff (bottom) concept distances for seen and unseen games across both datasets. Results from 5 runs plotted, with averages in bold.

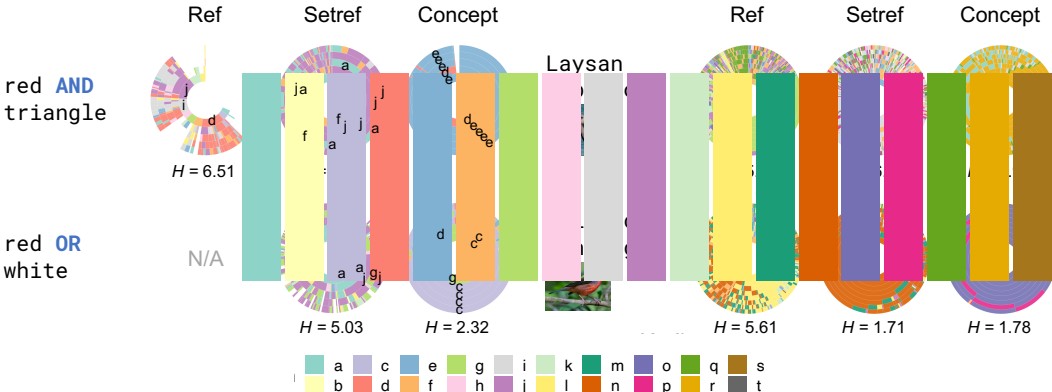

Figure 4: Distribution of 300 messages and entropies for game concepts in ref, setref, and concept settings. Messages start at the center and proceed outwards, with each colored section corresponding to a unique token and its frequency of occurrence at that position in the message. Empty spaces indicate end of sentence. For convenience, ShapeWorld plots are partially labeled with tokens.

## 6.2 Generalizing across game types

To measure the generality of the agents' strategies, we evaluate their ability to generalize zero-shot across game types. Table 2 shows accuracy and systematicity metrics for agents evaluated on games of a different type. Agents trained on the setref and concept games are able to generalize to reference games (yellow cells), producing systematic referring expressions, as they have already been biased towards generic language. Importantly, setref game agents can generalize to concept games with separate inputs (magenta cells), though to a lesser extent for Birds (75% vs 84%). This suggests that sets pressure agents to learn generalizable features *despite* the shared input. In contrast, we see little generalization ability from reference games to setref and concept games (orange cells), suggesting that agents are not conveying generalizable features, but rather spurious patterns in the input [3].

## 7 Probing for compositionality

The richer set of concepts afforded by our setref and concept games allow for more detailed analyses of the emergent languages beyond the standard metrics presented above. Broadly, we are interested in whether the compositional structure of concepts is reflected in the language. For example, our agents produce messages for the primitive concepts *red* and *triangle*, as well as the conjunctive concept *red AND triangle*.[6] Natural language is equipped with a composition operator, $\text{AND}(m_1, m_2) = m_1 \text{ AND } m_2$, that operates solely on lexical forms and whose meaning is defined as the conjunction of the meanings of its arguments. Does a similar operator exist in the emergent language (Figure 6a)?

---

[6]We cannot do this analysis for reference games, since we cannot test primitive concepts; recall Section 4.

Table 2: Accuracy/AMI/Topographic $\rho$ (concept Edit distance) for agents *trained* on different game types (columns), then *evaluated* (zero-shot) on different game types (rows). Chance accuracy is 50%. Gray shaded cells indicate standard test-time evaluation; other cell colors are explained in the text. Note that for ShapeWorld reference agents, we evaluate only on setref and concept games that use the 30 conjunctive concepts tested in reference games (e.g. *red triangle*, *blue square*).

|  | Train Ref | Train Setref | Train Concept |
|---|---|---|---|
| **ShapeWorld** | | | |
| Eval Ref | 98 (0.4)/.04 (.00)/.00 (.00) | 91 (4.3)/.43 (.09)/.42 (.11) | 83 (5.6)/.60 (.04)/.63 (.08) |
| Eval Setref | 56 (0.1)/.02 (.00)/.00 (.00) | 90 (1.3)/.59 (.08)/.12 (.03) | 83 (6.7)/.66 (.07)/.09 (.01) |
| Eval Concept | 50 (0.0)/.02 (.00)/.00 (.00) | 90 (4.6)/.59 (.08)/.12 (.03) | 82 (3.0)/.66 (.07)/.09 (.01) |
| **Birds** | | | |
| Eval Ref | 91 (0.8)/.05 (.00)/.04 (.01) | 85 (1.1)/.14 (.02)/.16 (.02) | 82 (0.9)/.20 (.02)/.15 (.01) |
| Eval Setref | 64 (1.7)/.03 (.01)/.15 (.01) | 84 (1.1)/.17 (.02)/.37 (.04) | 78 (0.7)/.26 (.02)/.40 (.03) |
| Eval Concept | 56 (0.9)/.03 (.01)/.15 (.01) | 75 (0.9)/.17 (.02)/.37 (.04) | 82 (0.6)/.26 (.02)/.40 (.03) |

We propose to *learn* such an operator by training a model to compose messages in the emergent language to form their conjunctions. Like the English AND, this operator must be able to combine any of the concepts in the language, and must crucially *generalize* to novel combinations of features. If, given new concepts, we can reconstruct a message that induces the right behavior in the student, this suggests our model has learned some analog of AND in the language. The reconstruction accuracy of this model can then be interpreted as a much more explicit measure of compositionality than the measures explored above: it reveals the degree to which the syntax of the emergent language operationalizes the specific logical operations present in the underlying space of concepts.

Our method is inspired by the *Tree Reconstruction Error* (TRE) metric proposed by Andreas [1], which learns a compositional approximation to a representation space, assuming the latent compositional structure is known.[7] However, there are several crucial differences in our method that are optimized for probing for linguistic structure. First, we *learn* a set of arbitrary composition operations, instead of imposing a predefined operation (e.g. elementwise addition). Moreover, these learned composition operators are actually valid and interpretable linguistic transformations on messages, rather than operations (like addition) that work only on internal representations of sub-concepts. And finally, we evaluate our operators on held-out concepts, examining how the reconstructed messages serve the ultimate communicative goal: inducing the correct generalization behavior in the student.

## 7.1 Learning an Approximate Compositional Reconstruction (ACRe)

Let us first formalize the learning problem: after training our agents, we have a dataset of message and concept pairs $\mathcal{T} = \{(m_i, c_i)\}$ generated by a teacher for each game. Each concept is one of a set of logical forms $\mathcal{L}(\mathcal{C})$ defined inductively over a set of *primitive* concepts $\mathcal{C}$ (e.g. *red*, *triangle*) and *composition operations* $\Omega$ as follows:

1. Every primitive concept is in $\mathcal{L}(\mathcal{C})$: $\mathcal{C} \subseteq \mathcal{L}(\mathcal{C})$.

2. Every composition of concepts is a concept: let $\Omega_n$ be the set of $n$-ary composition operations. Then $\forall n,\ (c_1, c_2, \ldots, c_n) \in \mathcal{L}(\mathcal{C})^n,\ \omega \in \Omega_n$, we have $\omega(c_1, c_2, \ldots, c_n) \in \mathcal{L}(\mathcal{C})$.

$\mathcal{T}$ defines a probability distribution over messages given concepts: $p_{\mathcal{T}}(m \mid c)$. Our aim is to learn an **A**pproximate **C**ompositional **Re**construction to these messages $\hat{p}_\eta(m \mid c)$, composed of $\eta$-parameterized message distributions that factorize along the compositional structure of $\mathcal{L}(C)$:

$$\hat{p}_\eta(m \mid c) = \begin{cases} \hat{p}_\eta^c(m) & \text{if } c \in \mathcal{C}, \text{ i.e. } c \text{ is primitive} \\ \mathbb{E}_{\hat{m}_i \sim \hat{p}_\eta(m_i \mid c_i)} \left[ \hat{p}_\eta^\omega(m \mid \hat{m}_1, \ldots, \hat{m}_n) \right] & \text{if } c = \omega(c_1, \ldots, c_n), \end{cases} \quad (3)$$

---

[7]We cannot apply TRE directly to our setting. Andreas [1] applied TRE to a standard reference game, where targets are represented as conjunctions of shape and color features (e.g. *blue square*, *red triangle*). As we mention in Section 4, because reference games cannot test primitive concepts like *red* and *triangle*, Andreas [1] proposed to *learn* representations for primitives which can then be composed via some (predefined) composition function. However, in our setting, it makes little sense to learn arbitrary representations for primitive concepts, when we actually have real messages for such concepts in the first place, hence the method we propose.

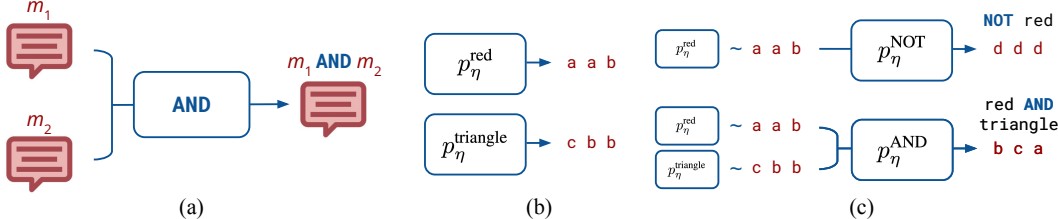

(a)               (b)               (c)

Figure 6: Our ACRe procedure. (a) Does a lexical analog of AND exist in our emergent language? (b) We first train primitive LMs to mimic the distribution of agent messages given a fixed concept. (c) We then train composition operations by sampling arguments from primitive LMs, then training a seq2seq model to mimic the agent message produced for a higher-order concept.

where $\hat{p}^c_\eta(m)$ is a model of the distribution of messages for primitive concept $c$, and $\hat{p}^\omega_\eta(m \mid m_1, \ldots, m_n)$ is a learned lexical analog to operation $\omega$ that takes in messages and outputs a distribution over composed messages. Given a concept $c$, we can sample a message from $\hat{p}_\eta(m \mid c)$ by either sampling directly from $\hat{p}^c_\eta$ (if $c$ is primitive) or recursively sampling messages $m_i$ from the constituents of $c$, then sampling from the corresponding $\hat{p}^\omega_\eta$.

We implement $\hat{p}^c_\eta$ and $\hat{p}^\omega_\eta$ as small 2-layer transformer-based language models (LMs) [38]. For each $c$, $\hat{p}^c_\eta$ is an unconditional LM (i.e. we have an LM for *red*, *triangle*, etc.). For $n$-ary operations $\omega$, $\hat{p}^\omega_\eta$ is a sequence-to-sequence (seq2seq) model decoding from the concatenated arguments: $\hat{p}^\omega_\eta(m \mid m_1, \ldots, m_n) = p^{\text{decode}}_\eta(m \mid m_1 \; \texttt{[SEP]} \; m_2 \ldots \texttt{[SEP]} \; m_n)$. Note that this formulation imposes few constraints on the mechanism of the composition operator: for example, we are not enforcing that there exists a token denoting conjunction (AND), or that the arguments must be copied verbatim into the output. These constraints, generally true of human languages, could be explored in future work.

To train ACRe, let $\mathcal{T}_c = \{(m_i, c_i) \in \mathcal{T} \mid c_i = c\}$ be the set of message-concept pairs with concept $c$, and $\mathcal{T}_\omega = \{(m_i, c_i) \in \mathcal{T} \mid c_i = \omega(\cdot)\}$ be the set of messages where $c_i$ uses $\omega$. Now, for each primitive $c \in \mathcal{C}$, train the LM $\hat{p}^c_\eta$ on $\mathcal{T}_c$ to approximate $p_\mathcal{T}(m \mid c)$. Then, freezing these models, for $n = 1, \ldots, N$, $\omega \in \Omega_n$, train the composition model $\hat{p}^\omega_\eta$ on $\mathcal{T}_\omega$. Specifically, given a pair $(m, \omega(c_1, \ldots, c_n))$, first sample messages for the sub-concepts $c_i$ from our frozen models: $\hat{m}_i \sim \hat{p}_\eta(m_i \mid c_i)$.[8] Then, train the composition model to maximize $\hat{p}^\omega_\eta(m \mid \hat{m}_1, \ldots, \hat{m}_n)$. For example, given a message $\texttt{bca}$ for the concept *red AND triangle*, we first sample messages for *red* and *triangle* from $\hat{p}^{\text{red}}_\eta$ and $\hat{p}^{\text{triangle}}_\eta$, then train $\hat{p}^{\text{AND}}_\eta$ to decode $\texttt{bca}$ from these messages (Figure 6). Full training and model details are in Appendix G.

## 7.2   Evaluation and results

After training, each $\hat{p}^\omega_\eta$ is a learned lexical analog to the composition operation $\omega$. To test whether our approximation has learned the semantics of the language, we hold out 20% of conjunctive and disjunctive concepts $c'_i$ during training and sample messages for these concepts from the learned $\hat{p}_\eta(m \mid c'_i)$ according to Equation 3. We evaluate how well these reconstructed messages encode the concepts via (1) lexical overlap with the true teacher messages (as measured by BLEU) and (2) student performance when given our language and a game for the corresponding concept.[9]

Results are in Table 3. The upper bound on performance is given by the true **Teacher** language, $p_\mathcal{T}(m \mid c)$. We also use language sampled randomly from (1) teacher messages for *any* concept $p_\mathcal{T}(m)$ (**Random**) and (2) teacher messages for the **Closest** concept as measured by Edit distance (breaking ties randomly). ACRe's performance is a measure of the degree of compositionality in the language, upper bounded by the teacher language and lower bounded by the random baseline. The results reveal some degree of compositional structure: ACRe reconstructs the training data well and crucially outperforms baselines in predicting messages for unseen concepts. Of course, our ACRe

---

[8]As presented, this procedure is only possible if the composition models can be trained and frozen in a specific order without circular dependencies. For example, we cannot naively train $\hat{p}^\omega_\eta$ on concepts of the form $\omega(\omega(\cdot))$, since sampling from the inner $\omega$ requires an already-trained $\hat{p}^\omega_\eta$. Learning from arbitrary concepts is possible by backpropagating through samples (e.g. via Gumbel-Softmax), which we leave for future work.

[9]In these experiments, we run ACRe on agents trained on the full set of 312 concepts in ShapeWorld, since we are not testing compositional generalization of the agents, but of ACRe.

Table 3: ACRe evaluation. (SD) across 5 runs. In the highlighted cells we conduct paired $t$-tests, comparing ACRe to Closest; * indicates significance at $p < 0.05$.

| Game | Language | Train | | | Test | | |
| --- | --- | --- | --- | --- | --- | --- | --- |
| | | BLEU-1 | BLEU-4 | Student Acc | BLEU-1 | BLEU-4 | Student Acc |
| **Setref** | Teacher | 100 (0.0) | 100 (0.0) | 91 (2.7) | 100 (0.0) | 100 (0.0) | 86 (5.2) |
| | ACRe | 96 (2.1) | 73 (5.0) | 81 (3.4) | 91 (3.9)* | 52 (10.0)* | 65 (3.4)* |
| | Closest | 78 (4.6) | 28 (6.2) | 48 (0.7) | 88 (3.8) | 38 (9.0) | 56 (0.7) |
| | Random | 71 (6.6) | 22 (4.3) | 50 (0.0) | 73 (7.4) | 24 (5.5) | 50 (0.3) |
| **Concept** | Teacher | 100 (0.0) | 100 (0.0) | 88 (4.2) | 100 (0.0) | 100 (0.0) | 84 (4.4) |
| | ACRe | 95 (3.4) | 80 (7.6) | 82 (2.2) | 87 (6.9)* | 59 (12.2)* | 70 (4.0)* |
| | Closest | 64 (3.7) | 28 (6.2) | 48 (1.6) | 76 (3.2) | 38 (6.7) | 56 (1.3) |
| | Random | 54 (3.0) | 19 (2.8) | 50 (0.2) | 56 (2.2) | 20 (3.2) | 50 (0.4) |

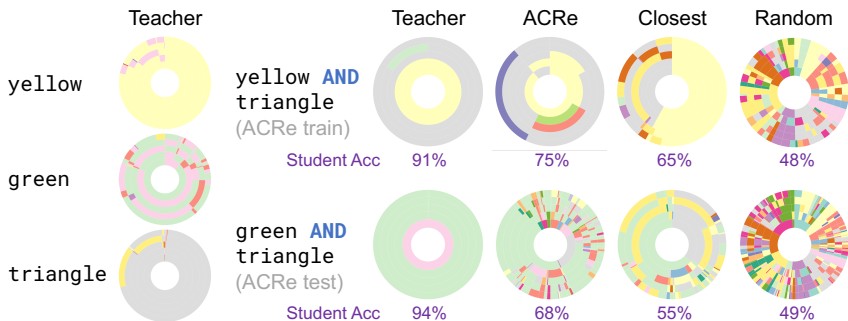

Figure 7: Composition in the emergent languages. A concept game teacher's messages for primitive concepts *yellow*, *green*, and *triangle*, conjunctions *yellow AND triangle* and *green AND triangle* (where *green AND triangle* is an unseen combination for ACRe), and predicted messages according to ACRe and other baselines. Color key same as Figure 4.

model trails Teacher language accuracy by 16–20 points. This gap could stem from a failure of ACRe to find the correct compositional generalization, or lack of compositionality in the language itself.

A qualitative analysis supports the interpretation that the compositional operations are not always interpretable. Figure 7 shows an example of message distributions for the primitive concepts *yellow*, *green*, and *triangle*, as well as distributions for the conjunctive concepts *yellow AND triangle* and *green AND triangle*, with predictions from ACRe and baseline models. The message for *yellow AND triangle* is intuitively composed out of tokens concatenated from both primitives similar to natural language [12]. However, the message *green AND triangle* uses tokens from *green*, but none from *triangle*, thereby violating the *mutual exclusivity* principle of language [28]. Regardless, in both cases our ACRe model is able to approximate such messages better than the other baselines. An exciting avenue for future work is encouraging models to develop operators more akin to human language, and evaluating their acquisition by searching among a more restricted class of ACRe models $\hat{p}_\eta^\omega$.

## 8 Related Work

**Promoting compositionality in multi-agent communication.** Compositionality and systematicity in emergent languages have long been *a priori* goals in multi-agent communication. Such languages may be easier to interpret, teach, and integrate with human language via supervised learning [26]. Towards this end, a large body of work (see [20] for review) has explored what factors might encourage compositionality in emergent communication, such as teachability and iterated learning [25, 30], agent capacity [31] and regularization [27], self-understanding [8], and game design [16, 22]. However, most of this existing work operates within the limited paradigm of the Lewis [24] reference game. In this paper, we propose to revisit and revise the fundamentally limited reference game objective: inspired by human teaching [9] and generic language [36], we encourage our agents to communicate *generalizations* over objects, which significantly increases linguistic systematicity, orthogonal to any of the alternative pressures proposed in related work.

**Measures of compositionality in languages.** Crucial to the problem of promoting compositionality in emergent communication is how to measure it in the first place. The literature has seen a wide variety of methods [1, 2, 4, 6, 31] claiming to more accurately align with human notions of compositionality, some of which are reported here. Most of this existing work focuses on outputting a broad scalar quantity that represents the degree of compositionality in a language [e.g. topographic $\rho$; 4]. In contrast, ACRe is a much more granular attempt at measuring not just *how much* compositionality, but *what kinds* of compositionality emerge, by actually learning, evaluating, and interpreting each distinct compositional operation.

This brings our work more in line with more precise studies of the emergence of composition operations in emergent languages [34, 35] and the analyses of van der Wal *et al.* [37] and Andreas [1]. In contrast to Steinert-Threlkeld [34, 35], who studies simpler settings where compositionality can be verified with manual inspection, we propose a way to measure compositionality in more complex languages that clearly do not exhibit perfect compositionality, but may still have learnable latent structure. In contrast to van der Wal *et al.* [37], who use grammar induction techniques for syntactic analysis of emergent languages, we tailor our syntactic analysis to messages generated for a known semantic space of concepts. This lets us approximate concrete syntactic operations in the language, and evaluate how well the approximations capture the corresponding semantic operations. Lastly, our method ACRe builds off of the TRE metric developed by Andreas [1], and we describe this relationship in Section 7.

**ACRe as program induction and grammatical inference.** Finally, ACRe is reminiscent of a program induction or grammatical inference problem, where inputs are agent messages for primitive concepts, and outputs are the messages produced after some composition has been applied to the inputs. Our task is to discover the (ideally simple and interpretable) lexical programs that implement the corresponding transformation in semantic space. Because we have no priors over what lexical transformations, if any, the emergent languages might implement, we search for programs among a general class of seq2seq translation models. However, in this domain, human languages have much simpler lexical operators involving concatenation and infix notation (e.g. `x AND y`), and in the future, we would like to push emergent languages towards stricter compositionality. One way of benchmarking more cleanly compositional languages is to restrict ACRe models to more constrained and interpretable programs learned with techniques from the program synthesis [11] or grammar induction [10] literature.

## 9   Conclusion

We have proposed extensions of referential games to sets of objects, and found that the need to convey generalizable categories leads to the development of more systematic languages, whether inputs are shared (setref) or unshared (concept) across agents. Moving forward, the richer space of concepts afforded by our setref and concept games are a promising testbed for studying the emergence of higher-level linguistic phenomena, such as quantifiers or probabilistic language. Finally, while ACRe reveals some compositional structure in the emergent languages as-is, the learned composition operations are not particularly interpretable. Future work should identify what kinds of environmental or architectural constraints might encourage more transparent composition in learned artificial languages. One challenging evaluation of compositionality along these lines is to measure the ability of agents to extrapolate to longer and more complex concepts not seen during training (e.g. *green OR (blue AND triangle)*), and evaluating ACRe's ability to capture this recursive structure.

## 10   Broader Impact

Our work investigates agents communicating in artificial and isolated environments, and thus has limited immediate societal impact. However, we can imagine that with future advances in research and compute, agents may learn linguistic strategies to collaborate on real-world problems in increasingly high-stakes domains, and it will be important to ensure that the learned languages are interpretable, safe, and reliable. Our work has potential positive benefits in these scenarios: our proposed games encourage agents to learn more human-like linguistic behavior, which might ease collaboration with humans; and ACRe is a tool for evaluating and interpreting learned languages. However, future work is needed to see whether these tools remain reliable as we scale up our agents and tasks.

## Acknowledgments and Disclosure of Funding

We thank Alex Tamkin, Elisa Kreiss, Josh Rozner, Gabriel Poesia, and Chris Potts for helpful comments and discussions, and our anonymous reviewers for extensive feedback on the paper. This research was supported by an NSF Graduate Research Fellowship for JM, the Stanford HAI-AWS Cloud Credits for Research program, and the Office of Naval Research grant ONR MURI N00014-16-1-2007.

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
