# A Model, training, and dataset details

All models are trained end-to-end with the Gumbel-Softmax [14] trick with the Adam [15] optimizer with learning rate 0.0001. Models are trained on a single Titan Xp GPU on an internal cluster. Training time is typically 6-8 hours on 4 CPUs and 32GB of RAM. Code and data are available at `https://github.com/jayelm/emergent-generalization`.

## A.1 ShapeWorld

**Model.** $f_\theta^T$ and $f_\phi^S$ are 4-layer convolutional neural networks, each consisting of a 64-filter 3x3 convolution, batch normalization, ReLU nonlinearity, and 2x2 max-pooling layer, as used in the few-shot learning literature [33]. RNN encoders and decoders are single layer Gated Recurrent Units (GRUs) [7] with hidden size 1024 and embedding size 500. We train with batch size $B = 128$.

We noticed that for ShapeWorld specifically, our setref and concept games easily converged to local minima with approximately 83% maximum accuracy by only considering color features and ignoring shape features. In these experiments, we had speakers sample tokens from a mixture of 90% the original logits, and 10% a uniform distribution over tokens, to encourage exploration (similar to $\varepsilon$-greedy policies in RL), which improved performance across all games.

**Data.** As aforementioned, for reference games there are 30 concepts (conjunctions of shape and color e.g. *blue square*, *green ellipse*), and for setref and concept games there are 312 total concepts (see Appendix B), of which 80% are reserved for training and 20% are reserved for test. From the training concepts, we sample 20,000 base games to use as our training set, each with 40 targets and distractors each. At training time, we perform augmentation by randomly selecting 10 targets and 10 distractors given to both teacher and student, meaning that the total set of games is combinatorially large (over $\binom{50}{10}$ combinations for reference games, and $\binom{50}{10}^2$ combinations for setref and concept). We set up validation and test datasets with 2000 games each, divided among seen and unseen concepts, with no augmentation performed. We train over 100 epochs (defined by a single pass through 20,000 augmented games) until average performance on the validation set is maximized.

**License.** ShapeWorld is distributed under an MIT license (`https://github.com/AlexKuhnle/ShapeWorld/blob/master/LICENSE`).

## A.2 Birds

**Model.** $f_\theta^T$ and $f_\phi^S$ is an ImageNet [32]-pretrained ResNet-18 [13]; similar results were observed for models trained from scratch. Like ShapeWorld, RNN encoders and decoders are single layer GRUs with hidden size 1024 and embedding size 500. We train with batch size $B = 16$ and preprocess images with ImageNet mean normalization.

**Data.** From the 100 training classes, we sample games dynamically by randomly selecting 5 positive targets from the class and 5 negative targets randomly. Like in ShapeWorld, this makes the number of possible training games combinatorially large. We set up validation and test datasets with 400 games each divided among seen and unseen concepts. We define an epoch as a single pass through 1,000 augmented training games, and like before, select the model with the highest performance on the validation set after 100 epochs.

For additional example games from both datasets, see Figure S1.

**License.** Caltech-UCSD Birds dataset is not distributed under a license; images are retrieved from Flickr and are the property of the respective photographers (`http://www.vision.caltech.edu/visipedia/CUB-200-2011.html`).

# B ShapeWorld concepts

The 312 ShapeWorld concepts are either:

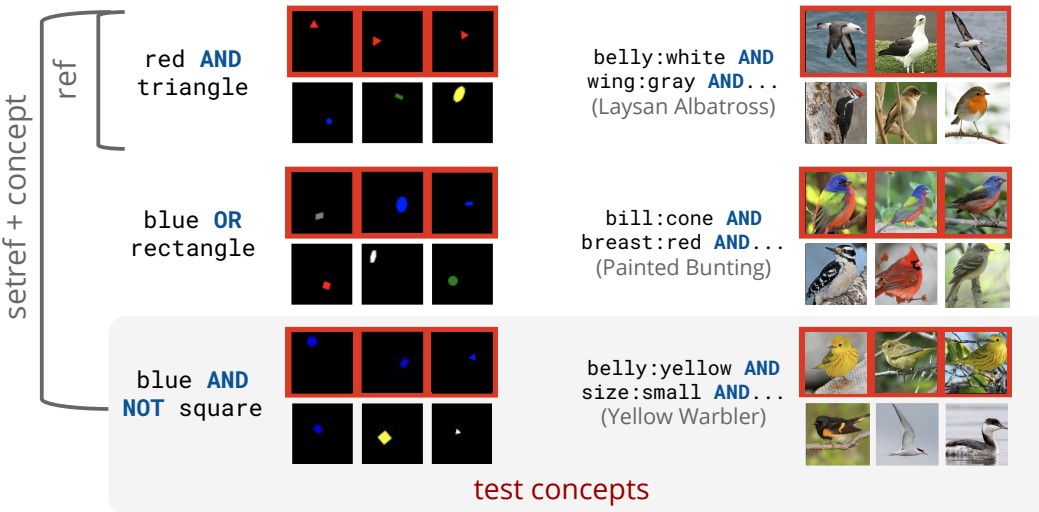

Figure S1: More example games for both ShapeWorld and Birds datasets.

1. A single *primitive* shape (*triangle, square, circle, ellipse, rectangle*) or color (*red, blue, green, yellow, white, gray*), possibly negated (e.g. *not gray*);

2. A disjunction of two (possibly negated) primitives (e.g. *blue or yellow*, *circle or not red*);

3. A conjunction of two (possibly negated) primitives (e.g. *red and triangle*, *red and not triangle*).

We enumerate all (boolean-equivalent) possible formulas, then discard formulas which are tautologically true (e.g. *not yellow or not red*) or unsatisfiable (e.g. *circle and square*).

For each concept, sampling positive and negative shapes uniformly often results in games that do not specifically test the concept. For example, for the concept *gray and not circle*, there may not be any negative *gray circles*, so the agent could just infer the concept *gray*. To ensure that concepts are fully tested, for disjunctive concepts, we sample 1/3 targets that satisfy *only* the *left* side of the disjunction; 1/3 that satisfy only the *right* side; and 1/3 that satisfy both. For conjunctions, we sample 1/3 *distractors* that only fail to satisfy the left side of the disjunction; 1/3 that only fail to satisfy the right side; and 1/3 that fail to satisfy both sides.

Code used for generating the dataset is available at `https://github.com/jayelm/minishapeworld/tree/neurips2021`.

## C  Varying communication channel size

To see whether our results hold as we vary the bandwidth of the communication channel, we run agents on both datasets with the following configurations of (vocabulary size, max message length):

- **Small** (S): (3, 3), leading to only 27 possible messages, which is not enough messages to uniquely identify each concept in any of the games explored here
- **Medium** (M): (5, 5)
- **Large** (L): (100, 20)
- **X-Large** (XL): (1000, 20)

Results are in Figure S2. Our conclusions are as follows:

1. Training for concept games is less stable, and with very small (Shapeworld S) and very large (Birds XL) vocabulary sizes is often unable to learn.

2. Outside of the concept game agent failures, accuracy on both seen and unseen concepts tends to increase as we increase the communication channel size.

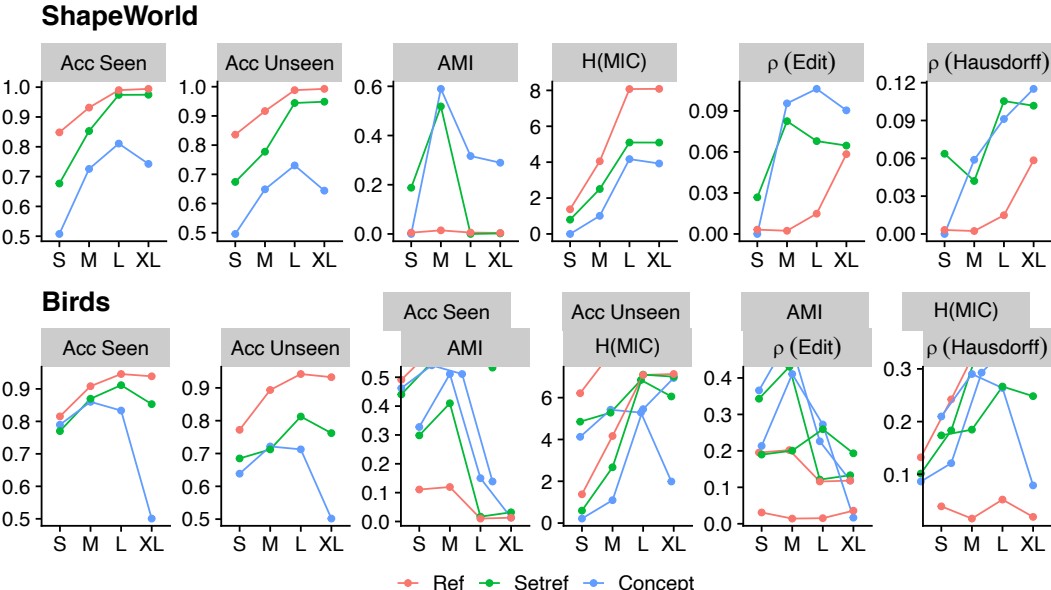

Figure S2: Accuracy and language systematicity metrics while varying the communication channel bandwidth for both datasets. Each dot represents one run.

3. Across all games, the information theoretic measures tend to show less systematicity as we increase the communication bandwidth. This is likely because as the number of vocabulary tokens and message length increases, the chance of sampling an errant token while generating a message increases, which is then treated as a completely unique message under our information theoretic measures; thus some increased entropy and reduced AMI is expected.

    (a) Regardless, when comparing between games with equal channel sizes, setref and concept games generally have more systematic language. Concept games are more consistently systematic than reference games, except for the degenerate settings where it is unable to learn. Setref games are more systematic than reference games, except as measured by AMI in large vocabulary spaces (L, XL). The differences across the game types are smaller when the channel size is very large, e.g. in Birds L and XL.

4. Similarly, across all channel sizes, topographic $\rho$ tends to be higher for setref and concept games. Somewhat mysteriously, the topographic $\rho$ measures do not respond to varying channel sizes as the information theoretic measures do. In fact, for ShapeWorld, increasing channel size *increases* topographic $\rho$, especially for reference games. More work is needed to identify the sources of this effect.

To summarize: for communication channel sizes reasonably sized according to the dataset (e.g. the medium setting here, and the setting reported in the main text), our conclusions hold; for extremely small or large communication channel sizes, our conclusions still hold, but care must be taken to ensure stable training in these regimes, especially for concept game agents.

## D   Upper bounds on listener performance for setref and concept games

One way of calibrating the quality of emergent languages developed by our agents in setref and concept settings is to evaluate a listener on the set classification task given an "ideal" language. While we do not have human data generated for these games, we can use proxies to obtain upper bounds on listener performance. For ShapeWorld, we give the listener the ground-truth concept descriptions; for Birds, we use a randomly sampled caption associated with one of the images in the target class, using the language corpus collected by Reed *et al.* [29]. Table S1 shows results. With the true ShapeWorld concepts, listeners are able to attain near-perfect performance, suggesting that the agents in our settings have an imperfect understanding of the true concepts in each game. However, for Birds performance, emergent language performance is actually comparable to the ground-truth language performance (at 79% and 71%) performance on seen and unseen tasks, respectively). This

suggests that the emergent languages for this task do quite well, even reaching the upper bounds on performance for the agent architectures we explore in this paper.

Table S1: Performance of listener agents on the ground-truth setref/concept task when given ideal (human) languages. Note setref and concept are the same, since there is no teacher input.

| Dataset | Acc (Seen) | Acc (Unseen) |
|---|---|---|
| ShapeWorld | 99.8 (0.1) | 99.8 (0.1) |
| Birds | 79.3 (0.4) | 70.6 (2.0) |

## E   Experiments with traditional cross-entropy reference games

We presented an atypical formulation of reference games as consisting of multiple targets, with student decisions made independently:

$$p^S(Y^S \mid X^S, m) = \prod_i p^S(y_i^S \mid x_i^S, m), \tag{4}$$

where students are trained with the binary cross entropy loss in Equation 1, restated here for convenience:

$$\mathcal{L}_{\text{BCE}}(T, S, G) = -\sum_i \log p^S(y_i^S \mid x_i^S, \hat{m}), \quad \hat{m} \sim p^T(m \mid X^T, Y^T). \tag{5}$$

This was done to keep training objectives and models identical across games, and to keep the amount of training data consistent (i.e. there are exactly the same number of targets and distractors seen by each agent across training).

However, the typical reference game has a single target: instead of $Y^S \in \{0, 1\}^n$, we have a single target $t^S \in [1, n]$ denoting the index of the single positive example. Then the student probability that input $i$ is the target is the softmax-normalized

$$p^S(i \mid X^S, m) = \frac{\exp(\text{RNN-ENCODE}(m) \cdot f_\phi^S(x_i^S))}{\sum_{i'} \exp(\text{RNN-ENCODE}(m) \cdot f_\phi^S(x_{i'}^S))}$$

and the training objective for a single game is

$$\mathcal{L}_{\text{XENT}}(T, S, G) = -\log p^S(t^S \mid x_i^S, \hat{m}), \quad \hat{m} \sim p^T(m \mid X^T, Y^T). \tag{6}$$

To ensure that our alternative formulation did not affect results, we ran 5 experiments with the standard reference game trained with cross entropy loss, with a single target and 10 distractors. Figure S3 summarizes the relevant statistics; besides slightly higher topographic $\rho$ and AMI for the cross-entropy reference games for ShapeWorld, there are no qualitative differences compared to our reference game formulation and our conclusions are unchanged.

## F   Additional plots of speaker messages

See Figure S4 for additional plots of teacher messages made for ShapeWorld and Birds games. Overall, the plots show a general reduction in language complexity from ref to setref to concept, although some quirks emerge: for example, some characters (e.g. *e* in concept) appear to be overloaded (across *green ellipse* and *red*), and concept uses similar language for *painted bunting* and *yellow warbler*. White gaps indicate end of sentence, so there are games where the speaker teacher utters nothing (e.g. *blue or not circle* concept; *not red* setref).

## G   ACRe model, training, and dataset details

Like the agents, ACRe models are trained with the Adam optimizer and learning rate 0.0001 on a Titan Xp GPU. Training ACRe takes around 1-2 hours on 4 CPUs and 16GB of RAM.

**Models.** $\hat{p}^c_\eta$ is implemented as an unconditional Transformer [38] LM with 2 hidden layers, 2 attention heads, a word embedding size of 50, hidden and intermediate sizes of 100, and $\hat{p}^\omega_\eta$ has the same decoder, but also has a Transformer encoder with identical parameters and cross attention from the decoder to the encoder. The vocabulary of the Transformers are the vocabulary of the emergent language plus a special [SEP] token used to concatenate arguments for higher-order $\omega$ operations. These are implemented with Huggingface's Transformers library [41].

Concretely, this means that for ShapeWorld, we have 11 unconditional transformer LMs modeling each $\hat{p}^c_\eta$, and 3 transformer LMs, one modeling the unary operation NOT, and two modeling AND and OR.

**Data.** To train ACRe to approximate the language for a teacher-student pair, we sample 200000 messages from the teacher, evenly distributed across all games. These are then stratified into data used to train each primitive LM and data used to train each higher order operation. All LMs are trained with standard language modeling techniques with teacher forcing. We first train the primitive LMs on their respective data. Then we train the unary NOT model on concepts of the form NOT($c$) where $c$ is primitive, sampling a message for $c$ using the primitive models. Finally, we train the AND and OR models on the conjunctive and disjunctive concepts, sampling arguments from the primitive models—and the NOT model, if the argument is negated.

We train ACRe models for 20 epochs and do early stopping as follows: for the training of the primitives and the NOT model, we divide the agent messages into 90% training data and 10% data, and define one epoch as one pass through the training data, stopping training once performance on the validation data is maximized. For the training of the AND and OR models, since we would like to test compositional generalization, we divide the data *by concept*: we stratify all unique conjunctions and disjunctions into a train/val/test split of 80%, 10%, and 10% respectively. Models are trained on messages representing the 80% of training concepts (1 epoch = 1 pass through this data), with early stopping performed when performance is maximized on the messages generated for the unseen validation concepts. Finally, we evaluate on the final 10% of unseen test concepts. To produce the numbers presented in Table 3, we evaluate listener and BLEU-1 performance across 110000 ShapeWorld games and 7000 Birds games (i.e. 5 passes through the train and test data for each dataset), grouping the metrics by whether the concept belongs to either the train/val or test ACRe splits.

Note that we do not test generalization of the NOT operation as there are too few NOT operations to be able to generalize properly. We verify this by training our NOT model to attempt to predict the (perfectly compositional) *ground truth concepts*: in other words, the task is to predict *green* from *not green*, *blue* from *not blue*, and then use these to finally predict the negated version of *red*. However, the model completely fails to generalize as there are only 10 unique concepts, so the model

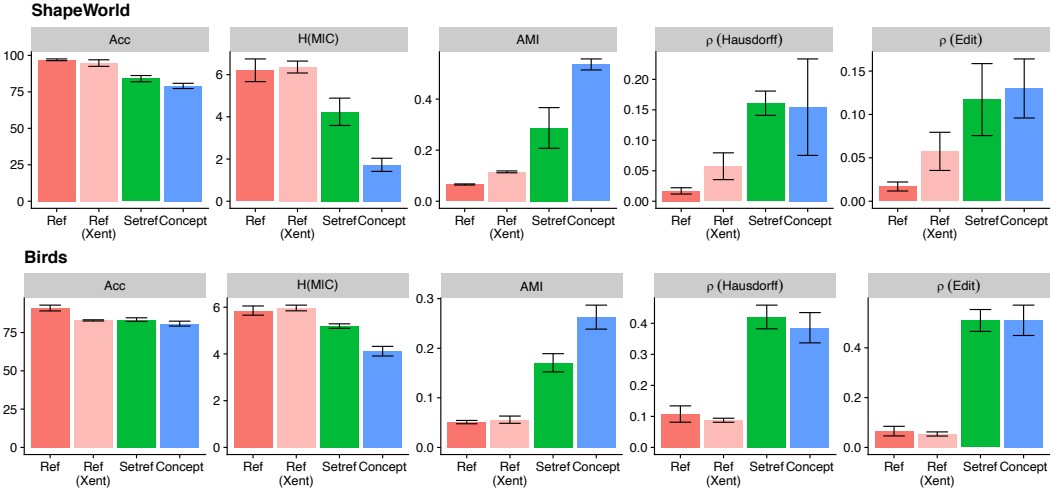

Figure S3: Accuracy and measures of language systematicity for reference games, setref games, and concept games, as well as reference games trained with the traditional cross entropy (xent) objective.

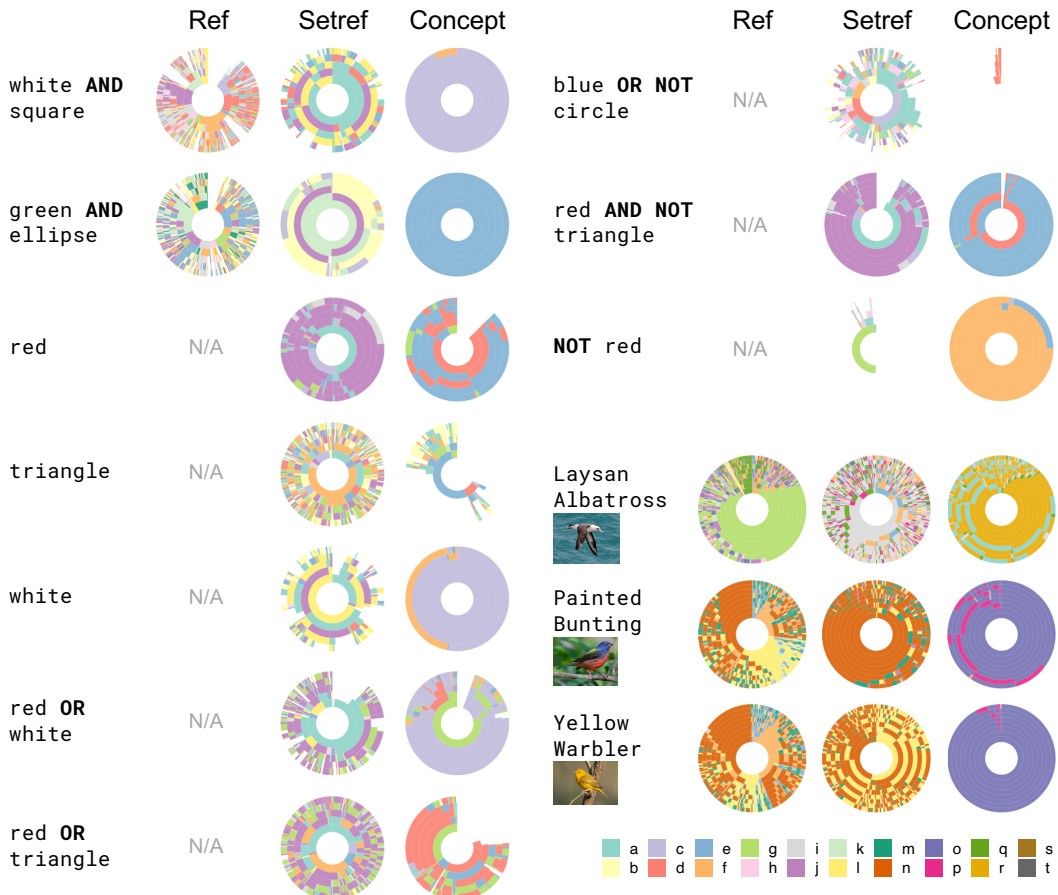

Figure S4: Additional plots of teacher messages for selected ShapeWorld and Birds games. Most ShapeWorld concepts are not tested in reference games, so those plots are not available.

simply memorizes the training set. Note that we still use the NOT model in sampling messages for conjunctive and disjunctive concepts that include NOT($c$) as an argument.