# OpenReview forum: "Emergent Communication of Generalizations"
_NeurIPS.cc/2021/Conference — NeurIPS 2021 Poster_

### Official Review · Reviewer_uEmD · 2021-07-15

**Rating:** 7
**Confidence:** 5

**Summary:**

This paper explores language emergence properties through two aspects:
 - extending the classic Lewis Discrimination Game to a multi-reference/concept one.
 - probing language compositionality with a new protocol called ACRe evaluation

Overall, it is a well-designed paper. It slowly introduces the concept and detail the intuition. I greatly appreciate the didactic aspects of the paper.
 Experiments look sounds, and there is a nice balance between quantitative and qualitative results.
Some visualizations -- without being ground-breaking -- are novel and help understanding communication protocols.

The paper does not have substantial weaknesses. Although the ideas are simple (they slightly extend classic ideas), they are correctly explored, and there is no overclaim.
My main concern may be that further work could have been done to bridge these ideas to the ML literature.
 Indeed, the Lewis Game extension is a form of negative sampling (which recently (re)boom with CPC)
ACRe is a naive approach for language automata.
As those ideas already in the literature, the contribution sometimes feels a but too incremental.
Therefore, correctly bridging those concepts would be an actual core contribution that is lacking in the community.

 In the end, I enjoy the paper, and it may nicely pile up in the emergent communication literature.
 Yet, it misses this je-ne-sais-quoi that could make it excellent.
 I thus lean toward acceptance (but not clear acceptance).

**Main Review:**

As mentioned, there are no apparent flaws in this paper, so I do not have strong remarks.

 A) Could you detail further the link between the game and different fields of the ML. For instance:
  - The two games can be seen as negative sampling approaches [1]
  - The reference set could be compared to weakly supervised learning (multiple-instance learning) [2-3]
 - How ACRe compare as a language automata [4-5]

Please, if you disagree with some of these comparisons, can you elaborate further? Note those are not extensive cites.

B) l70: the sentence with label $\hat{y}_iˆS$ creates a minor ambiguity as I thought It was a classification task for a few moments

C) l120 ”hypergeometric of randomness”… sounds clever, but it is pretty obscure :) Maybe something like a non-uniform distribution with some details about this non-uniformity would be more precise.

D) In evaluation, can you detail a bit further the accuracy metric. How many positive concepts, images do you have in average. Do you compute the mean accuracy or a success ratio (all concepts are found).

E) In tables,  please use floating numbers as the std accuracy is $10^-1$

F) Can you define further “systematic” communication. There is room for ambiguity.

G) When probing compositionality? Is there some specific reasons you did not implement TRE? Besides, why only computing BLEU1 as there are multiple characters in a message. Would you mind adding B2 at a bare minimum? Differently, 10% test hold may look a bit small. Did you try to have hard held-out composition (in the same spirit as [6])

[1] Kalantidis, Yannis, et al. "Hard negative mixing for contrastive learning." arXiv preprint arXiv:2010.01028 (2020).
[2] Song, Jiaming, and Stefano Ermon. "Multi-label contrastive predictive coding." arXiv preprint arXiv:2007.09852 (2020).
[3] Carbonneau, Marc-André, et al. "Multiple instance learning: A survey of problem characteristics and applications." Pattern Recognition 77 (2018): 329-353.
[4] Languages, Automata, and Logic
[5] Chapter 2 Languages and Automata - Inria
[6] Cogswell, Michael, et al. "Emergence of compositional language with deep generational transmission." arXiv preprint arXiv:1904.09067 (2019).



**Time Spent Reviewing:**

4h

---

> ### Author Response · Authors · 2021-08-10
> **Response to reviewer uEmD**
>
> Thank you for the detailed and thoughtful review!
>
> ## A) Connection with related work
>
> ### Set/concept reference games as negative sampling approaches
> We are aware of concurrent work drawing connections between standard Lewis-style reference games and contrastive learning, e.g. [Dessi et al., 2021](https://arxiv.org/abs/2106.04258), where authors view SimCLR as playing a reference game between two (identically-parameterized) models with a continuous communication channel. As reviewer uEmD points out, careful sampling of hard negatives has been shown to be beneficial in contrastive learning. Similarly, in emergent communication, negatives are often carefully sampled for reference games (e.g. [Lazaridou et al., 2018](https://arxiv.org/abs/1804.03984)) to encourage the agents to learn the right features—we depend on this in our work as well.
>
> We don’t believe, however, that our innovation has to do with negative sampling. Rather, we are changing the positive targets in a reference game from a singular target to a set. The choice of negative samples is still important: like in standard reference games, they should test understanding of the target concept in difficult ways. We describe our method for sampling negatives for ref, setref and concept games in Appendix B.
>
> We are thus not sure what reviewer uEmD means when they describe our work as incremental over SSL/negative sampling approaches: the connection between SSL and classic Lewis reference games has been established, and using hard negatives is standard procedure in reference games and is not limited to the setref and concept games we propose. Rather, we pursue novelty in an orthogonal direction: moving from single images to sets. We invite reviewer uEmD to clarify if we have misunderstood the connection.
>
> ### Comparison to multiple instance learning/set representation learning
> Thanks for proposing the connection to multiple instance learning (MIL). Indeed, in our games, the speaker’s task can be interpreted as MIL where the “witness rate” (i.e. proportion of positive examples in the bag) is 100%. The connection to weakly supervised MIL is less relevant, since it is concerned with tasks where each positive bag contains a small percent of positive examples. We could imagine extending our method to noisy/probabilistic concepts (e.g. not every instance in our target set is guaranteed to encode the concept), which would make weakly-supervised MIL methods more relevant.
>
> There is a more general connection to representation learning for sets ([Zaheer et al., 2017](https://arxiv.org/abs/1703.06114)): the goal of the speaker is to aggregate information across multiple instances into a coherent concept. In the same way that better set representation learning methods may enhance MIL (e.g. [Ilse et al., 2018](https://arxiv.org/abs/1802.04712)), better set representation learning may improve setref/concept performance. Our model’s prototypical network ([Snell et al., 2017](https://arxiv.org/abs/1703.05175)) is one simple way of aggregating information across sets, which in principle can learn any set function ([Zaheer et al., 2017](https://arxiv.org/abs/1703.06114)), though it is not the only choice. We did try a transformer encoder which aggregates instance representations into a set representation, analogous to Set Transformers ([Lee et al.,2019](https://arxiv.org/abs/1810.00825)), with the aim of better capturing interactions among individual instances. This approach did not give any tangible performance benefit, though there are likely other ways we might improve performance.
>
> Finally, we thank the reviewer for the clever connection to multi-label CPC, especially given the connection between contrastive learning and reference games above. Parallel to Song and Ermon’s extension of contrastive learning to multi-label contrastive learning, we extend reference games to multi-label reference games. It’s likely that there are some theoretical connections to draw here, but we leave this for future work.
>
> ### ACRe as language automata
>
> We offer some thoughts below about the connections here, and invite reviewer uEmD to elaborate if we have misunderstood anything.
>
> It is true that the process for learning individual components of ACRe models (i.e. fitting the unconditional/seq2seq models corresponding to primitives and higher-order concepts) can be replaced with any arbitrary program induction algorithm with optional constraints on the kinds of programs that can be learned. For example, instead of Transformers, we could use explicit algorithms for learning finite state machines/transducers from data ([Cichello and Kremer, 2003](https://www.jmlr.org/papers/volume4/cicchello03a/cicchello03a.pdf)), as used in the literature on grammar inference ([Wyard, 1993](https://citeseerx.ist.psu.edu/viewdoc/download?doi=10.1.1.90.5820&rep=rep1&type=pdf)) and program synthesis ([Gulwani, 2011](https://www.microsoft.com/en-us/research/wp-content/uploads/2016/12/popl11-synthesis.pdf)). However, because these algorithms are often brittle and hard to train, and because we have no priors on what kind of compositionality might emerge, we use Turing-complete ([Perez et al., 2019](https://arxiv.org/abs/1901.03429)) seq2seq architectures as the base ACRe components. As mentioned in Line 263, a more restricted class of ACRe models would test for stricter forms of compositional structure.
>
> We believe that the interpretation of ACRe as an extension of prior work on language automata is incorrect. Our work can be viewed as an *application* of language automata (or any program induction procedure) as one component in a broader framework that seeks to answer an entirely different question: the degree to which an unknown language has compositionality learnable by a prespecified class of models. Besides TRE, we are unaware of other work that applies modern neural methods to the problem of learning arbitrary syntactic operations from data, for use in analyzing language compositionality.
>
> We will make the connection to language automata (as a potential component in ACRe) clearer.
>
> ## B-E)
> Thanks for the suggestions; we will incorporate all suggested revisions. **D):** accuracy metric is mean accuracy, rather than success ratio, i.e. partial credit is given.
>
> ## F)
> We agree the definition of the term is ambiguous and will clarify. We define systematicity as having overall human-like consistency between meanings and messages: e.g. as measured by mutual information between messages and concepts, or correlation between similar messages and similar concepts (as measured by topographic similarity). We use both systematicity and compositionality because we believe compositionality makes a more forcing assertion that units of meaning are predictably composed to form larger meanings in a language.
>
> ## G1) Why not TRE?
>
> In Section 7 of the [original TRE paper](https://arxiv.org/abs/1902.07181), authors implement TRE to analyze compositionality of an emergent language in a reference game, with inputs (e.g. blue square, red triangle) represented as symbolic feature vectors. As we describe in Section 4 of our paper, reference games cannot test primitive concepts like “red” or “triangle”. TRE, however, depends on learning representations for primitives which can then be composed via some defined composition function. Thus, TRE proposes learning internal representations for primitives that are not actual valid expressions in the emergent language (represented as one-hot vectors), but rather can be arbitrary continuous representations (i.e. allowing for assigning fractional counts to word indices), so long as they can be added together to approximate the real message.
>
> In our setting, this approach seems wrong: it makes no sense to learn arbitrary representations for primitive concepts when we actually have real messages for the concepts in the first place. Thus, ACRe uses a curriculum, first learning a distribution for these primitive messages, then learning the composition operations to compose primitives together.
>
> Moreover, TRE restricts itself to investigating compositionality under a limited composition function (i.e. additive compositionality), which only works with the artificial, TRE-specific representations of primitives as “fractional” messages. Since we actually have messages for primitive concepts, we cannot “add” them together. Instead, we must find a composition operator in the language. We would like to do so while placing as few assumptions as possible on what this operator looks like, hence we train generic seq2seq models.
>
> These two innovations form the basis of ACRe.

---

> > ### Author Response · Authors · 2021-08-10
> > **Response to reviewer uEmD (continued)**
> >
> > ## G2) 10% test set is a bit small. Why not BLEU-2?
> >
> > We chose BLEU-1 since we may not have any reason to expect sequential information to be important in the agents’ learned communication protocols (since there is no explicit pressure to learn sequential information, except perhaps due to learning dynamics imposed by the RNNs of the agents).
> >
> > Nevertheless, we agree that higher-order BLEU would be informative. We increased the ACRe test set percentage to 20%, and reran BLEU, BLEU-2 and BLEU-4 across 5 ACRe runs and report the numbers below.
> >
> > For brevity, these numbers are on the ACRe Test split only:
> >
> > | Game Type | Language Type   | Listener Acc | BLEU-1      | BLEU-2      | BLEU-4       |
> > |-----------|-----------------|--------------|-------------|-------------|--------------|
> > | Setref    | ACRe            | 65.4* (3.4)  | 91.1* (3.9) | 82.7* (7.0) | 51.7* (9.9)  |
> > | Setref    | Closest Concept | 56.1 (6.6)   | 87.9 (3.8)  | 76.0 (7.5)  | 38.3 (8.9)   |
> > | Setref    | Random          | 49.8 (3.1)   | 73.4 (7.4)  | 55.3 (10.0) | 23.8 (5.5)   |
> > | Concept   | ACRe            | 70.4* (4.0)  | 87.0* (7.0) | 78.6* (9.9) | 59.3* (12.2) |
> > | Concept   | Closest Concept | 55.5 (1.3)   | 75.6 (3.2)  | 61.1 (4.0)  | 37.6 (6.7)   |
> > | Concept   | Random          | 50.1 (0.4)   | 53.8 (3.0)  | 36.9 (3.0)  | 19.9 (3.2)   |
> >
> > Where * indicates significance of ACRe over Closest at least at p < 0.05. Observe that BLEU-4 and BLEU-2  follow the same trend as BLEU-1, with perhaps even bigger differences between ACRe and Closest Concept for higher-order BLEU. Thanks for the suggestion! We will report full results in the paper with BLEU-1 and BLEU-4.
> >
> > ## G3) Harder hold-out sets?
> >
> > In our ShapeWorld dataset, there is no large variance between the length of the logical forms, so we cannot test harder generalization and/or extrapolation, though this is an interesting avenue for future work.

---

> > > ### Comment · Reviewer_uEmD · 2021-08-11
> > > **Thank you for the rebutal**
> > >
> > > I do appreciate this high quality rebuttal, and one of my main concern (bridge to other ML works) as been correctly and throughfully addressed (I now have more reading to do!). I encourage the authors to integrate some of their answer in the paper.
> > > As a result, I am increasing my score from weak-accept to accept.

---

### Official Review · Reviewer_Azs8 · 2021-07-18

**Rating:** 7
**Confidence:** 4

**Summary:**

In the referential game setup that is widely used in the emergent communication literature, this paper proposes expanding the target into a target set, and studies the effects of different choices for the distractor and target sets. The authors find that choosing suitable sets leads to more consistent communication protocols and higher topographic similarity between concept and message spaces, but at the same time causes a drop in communicative success.

The other contribution of the paper is a composition-based evaluation of the protocols that emerge from the set-based training. Small transformers are trained to mimic operators such as ‘AND’ and ‘NOT’. Evaluating the listener performance on the messages produced by those operator models shows a non-trivial communication performance, albeit significantly reduced relative to the original speaker models. The authors conclude that there is limited evidence for compositionality in the communication protocols based on this experiment.


**Limitations And Societal Impact:**

See main review; there is ample attention for and recognition of the limitations and impact.

**Main Review:**

The method presented here is very reminiscent of, and indeed seems to be inspired by, the prototype networks technique from Snell et al. (NeurIPS 2017). In contrast to Snell et al., the authors here use a discrete intermediate representation (the message) and inspect and analyse its properties. It is hard to compare the accuracies because the datasets are different, and although that is not the focus of the authors here, it would help to more clearly place this work in the context of the existing literature.

The authors’ focus is the systematicity of the communication protocol, and how that is affected by the support and distractor sets chosen. The main finding, loosely paraphrasable as ‘more systematic information in the support and distractor sets leads to more systematic communication’ is nice to see, if not altogether surprising (e.g. Snell et al. also observe that training with more variation in the distractor set leads to better classification performance, which can be attributed to the same effect).

The search for operators that combine elementary concepts into composite ones is interesting. I salute the authors for including that section even though it does not contain very clearly positive results - putting the idea into writing, suggesting the technique, and publishing a somewhat neutral finding, are helpful contributions.

The significant drops in communicative success with sets or concept sets are unfortunate. If more systematic communication does not lead to more accurate performance (also on unseen stimuli), it might not be something to strive for in agents, in spite of the human bias in favor. I do wonder to what extent the model architecture and the complexity of the datasets can be at fault here. More inspection of the effects that various choices for the support and distractor sets have on the communicative success would make the paper more relevant for future research. The main reason for wanting systematic communication is the expected benefit for generalization. If that benefit does not materialize, the systematicity of the communication protocol becomes less interesting altogether.

All in all, while I believe the paper contains a number of interesting contributions, a big question is left open, with significant implications for the relevance of the paper. The paper has lots of interesting analysis, and I would love to see the same rigor applied to the accuracy drop question.


**Time Spent Reviewing:**

4

---

> ### Author Response · Authors · 2021-08-10
> **Response to Reviewer Azs8**
>
> Thank you for the thoughtful and constructive review!
>
> ## The main reason for wanting systematic communication is the expected benefit for generalization
>
> *[and therefore, if our agents do not exhibit a benefit in generalization, systematicity (and thus our games) may not be useful targets to optimize towards].*
>
> Thanks to reviewer Azs8 for this excellent point which has caused us to think more critically about our work and its framing in the paper. We will divide our response into three parts + a conclusion.
>
> ### 1. Many benefits to systematicity outside of generalization
>
> Before we discuss generalization and systematicity, we would like to make the overarching point that **even if there is no clear benefit to generalization, there are still important intrinsic reasons to investigate and encourage systematic communication in agents.** Perhaps the most obvious reason is **interpretability**: we want our agents to learn interpetable and safe communication systems that are aligned with human values. Systematicity and interpretability also have very concrete pragmatic benefits: more interpretable languages have been shown to be easier to **teach**, [both to other agents](https://arxiv.org/abs/1906.02403) and [to other humans](https://cocosci.princeton.edu/tom/papers/IteratedLearningEvolutionLanguage.pdf) (for eventual human-in-the-loop collaboration). Keeping agents as human-like in their communication as possible may also be useful in preventing emergent communication protocols from diverging too far from human language (i.e. *semantic drift*), which is crucial for the sample efficiency of methods which [combine emergent communication self-play with supervised training](https://arxiv.org/abs/2002.01093). Our work follows a long line of work in this area that analyzes and encourages systematic communication as an *intrinsic good*, regardless of any perceived generalization benefit ([1](https://arxiv.org/abs/1906.02403), [2](https://arxiv.org/abs/1804.02341), [3](https://arxiv.org/abs/1804.03984), [4](https://openreview.net/forum?id=ZbXlSL_xwtA), [5](https://arxiv.org/abs/1705.11192), [6](https://arxiv.org/abs/2106.02067), etc).
>
> That being said: the relationship between systematicity and generalization is a highly nuanced one which we now realize we did not clarify sufficiently in the paper.
>
> ### 2. Incorrect to compare setref/concept accuracy to reference accuracy
>
> First, it is incorrect to compare accuracy achieved by reference and setref/concept agents (Table 1), and thus infer that setref and concept agents show a “drop in accuracy” and exhibit “less generalization ability”. This is because ref, setref, and concept are **fundamentally different tasks**. Reference game agents obtain high accuracy on their task, but high accuracy is *not* a measure of human-like generalization: the task is inherently flawed! The agents succeed not by learning generalizable features, but instead communicating spurious visual patterns in the input. This is a known phenomenon in the literature ([Bouchacort and Baroni, 2018](https://arxiv.org/abs/1808.10696)) and we reproduce the finding here, with Table 2 showing that reference game agents completely fail if different inputs are given to each agent (i.e. generalization to concept games). If reference game agents’ language was indeed generalizable (i.e. the messages really meant “red square” or “blue triangle”), then their messages should be at least somewhat robust to superficial differences in the input given to either agent, but this is not the case.
>
> In other words, the high accuracy on unseen reference games in Table 1 is a mirage: it is not a sign of systematic generalization, but rather communication of spurious features. If we restrict ourselves to measuring generalization along the kinds of dimensions that humans care about (i.e. do they learn reusable primitive concepts of shape/color?), our results show that **reference game agents show no generalization ability at all!** In contrast, it is the setref and concept game agents which have learned some degree of generalization: Table 2 illustrates that setref agents are robust to changes in teacher/student input, and that both concept and setref agents are able to generalize to standard reference games.
>
> ### 3. Experiments to establish upper bounds on setref/concept performance
>
> **If reference game agent accuracy is not the upper bound on performance for setref and concept games, what is?** While human data on these games would be the best upper bound, we will establish some positive control experiments—(A) listener model trained with ground-truth language, (B) continuous communication channel. These will show that setref/concept agents actually do quite well at learning an effective and systematic language, and that imperfect performance is at least partially due to accuracy limits of the models themselves.
>
> #### **A. Student trained with ground-truth human language**
>
>
> We trained 5 student models to accept ground truth (human) language and play the listener side of the setref/concept game (since we have no teacher, setref/concept are identical). This answers the question: if we had a perfect language (i.e. language humans would use on this task), how well would our models do?
>
> | Dataset    | Acc (Seen) | Acc (Unseen) |
> |------------|------------|--------------|
> | ShapeWorld | 99.8 (0.1) | 99.8 (0.1)   |
> | Birds      | 79.3 (0.4) | 70.6 (2.0)   |
>
> For ShapeWorld, human language is the ground truth formula; for Birds, we do not have language in the context of setref games. Instead, we have image captions for each bird, and randomly select a caption for one of the target birds to serve as human language. Such descriptions are very detailed ([Reed et al., 2016](https://arxiv.org/abs/1605.05395)), and language for birds in a class is quite similar, since they are of the same species.
>
> #### **B. Continuous communication channel**
>
> We ran 5 setref/concept agents **with a large, continuous communication channel**, where the teacher sends its entire learned prototype to the student (Line 68). We can alternatively think of this experiment as a single protonet doing few-shot learning.
>
> | Dataset    | Game Type | Acc (Seen) | Acc (Unseen) |
> |------------|-----------|------------|--------------|
> | ShapeWorld | Setref    | 99.2 (0.1) | 99.2 (0.1)   |
> | ShapeWorld | Concept   | 87.7 (0.3) | 83.8 (0.7)   |
> | Birds      | Setref    | 92.4 (0.1) | 87.8 (2.7)   |
> | Birds      | Concept   | 87.5 (3.0) | 78.6 (0.8)   |
>
>
> #### **C. Emergent language results reported in our paper (for convenience)**
>
> | Dataset    | Game Type | Acc (Seen) | Acc (Unseen) |
> |------------|-----------|------------|--------------|
> | ShapeWorld | Setref    | 92 (2.2)   | 87 (1.6)     |
> | ShapeWorld | Concept   | 88 (3.4)   | 75 (3.0)     |
> | Birds      | Setref    | 89 (0.2)   | 78 (0.2)     |
> | Birds      | Concept   | 88 (0.1)   | 73 (0.3)     |
>
> We make the following observations:
>
> First, the Birds task is inherently challenging, even with ground-truth language or high communication bandwidth. In fact, setref/concept agents do better with their own developed language than with human language which accurately describes the features of the target bird class (e.g. for concept: 88 vs 79.3; 73 vs 70.6)! Moreover, their languages achieve performance close to that of continuous communication (e.g. for concept: 88 vs 87.5; 73 vs 78.6).
>
> Second, for ShapeWorld, setref games are solvable (~100% accuracy) given either the true language or continuous communication. This indicates that for setref, the language the agents emerge is imperfect; still we believe 92-87% accuracy is still reasonably high in this setting. On the other hand, in the absence of ground-truth language, concept games are still more challenging, even with continuous communication: with continuous communication as the upper bound, concept agents achieve comparable performance (87.5 vs 88, 83.8 vs 75).
>
> ### 4. Conclusion
>
> While none of these experiments are perfect in establishing an upper bound (human data would be ideal), these are the numbers we should be comparing to, rather than reference game performance (where the task permits a trivial solution that achieves near-perfect performance). Indeed, our agent performance is much closer to that of these positive controls. These results show that the agents’ lower performance is partially due to inherent difficulty of the tasks themselves, at least given our current toolbox of neural models. Of course, we would expect this from any proposed benchmark.
>
> We agree that ceiling performance has not yet been reached on our task, especially with concept game agents generalizing to out-of-domain inputs. This suggests room for future modeling work, perhaps in incorporating more sophisticated representation learning methods as reviewer Azs8 mentions (also see reply to reviewer uEmD). Regardless, we hope this response has convinced the reviewer that:
>
> 1. the numbers in Table 1 *do not* indicate that setref and concept agents have poorer generalization than reference game agents, since we cannot really compare their absolute numbers across their games. In fact, reference game agents are “gaming the task”, designing spurious protocols that achieve high numbers but do not generalize in the ways humans would want them to (Table 2);
> 2. compared to our agents’ ability to recognize visual concepts from ideal languages and with large (continuous) communication channels, our agents do a reasonable job of inventing a systematic and generalizable language (though we have certainly not solved our proposed benchmarks); and
> 3. outside of this discussion entirely, there are several other reasons to desire systematic communication in agents *a priori*.
>
> We will update our paper with the additional baselines and a more nuanced discussion of these points, and invite additional questions from the reviewers to resolve any further points of confusion.

---

> > ### Author Response · Authors · 2021-08-10
> > **Response to Reviewer Azs8 (continued)**
> >
> > ## Relation to Snell et al. 2017 should be made more clear.
> >
> > Thanks for this point; this is correct. We agree with the reviewer that our focus is on a different problem than few-shot learning (i.e. multi-agent communication), but we do use Prototypical networks as one component in the Teacher model whose aim is to learn a good concept representation from individual instances. With that in mind, there are connections to set representation learning which we will make more clear in the paper. See our response to Reviewer uEmD for a more detailed discussion of this point.

---

> > ### Comment · Reviewer_Azs8 · 2021-09-01
> > **raising score to 7**
> >
> > Thank you for your extensive response, and please accept my apologies for being slow to get back.
> >
> > I take both of your points: about generalization between the different games being a stronger kind of generalization than within a game, and about there being other reasons to be interested in systematicity.
> >
> > I'm raising my score to a 7.

---

### Official Review · Reviewer_cuD1 · 2021-07-19

**Rating:** 6
**Confidence:** 3

**Summary:**

In this paper, the authors point the limitation of existing reference games and propose different games that require communicating generalizations over sets of objects. In the set reference game, a teacher must communicate to a student a group of objects belonging to a concept. The concept game is more challenging as each agent sees different examples of the concept. The authors also build a teacher model and a student model to play these games. They build two tasks to examine the languages developed for their proposed communication games.




**Limitations And Societal Impact:**

The contribution is limited. The authors propose several new reference games and evaluation metrics. However, the proposed games seem very simple, even the Concept game seems more challenging than the Setref and Ref games.

It is obvious if the task is about high-level concept prediction, the models tend to use fewer tokens and the tokens are more consistent. But it will lose the detailed information. If the task is about low-level concept prediction, the models tend to use more tokens and capture more details. These seem very obvious and I am not sure why designing such tasks is a big contribution. It is more convincing if the authors can perform their experiments and get corresponding conclusions on more challenging tasks or environments.

The compositional experiment is interesting, but again the concepts/messages/tokens are very limited. What if the concepts need to be described by a long message? What if the difference between the two concepts is very small? Given the powerful natural networks used in recent works, it might be good to have more challenging tasks or environments.

**Main Review:**

It is good that the authors propose a new reference game that requires communicating generalizations over sets of objects.

The paper is well written and easy to follow.

The authors propose several metrics to analyze the systematicity and interpretability of the learned languages.

The experiments are good. The authors test the accuracy, conditional entropy of messages given concepts, and adjusted mutual information score of different tasks.

It seems the tasks are easy and the concepts/words/language that appeared in different games are limited. It is hard to verify whether the conclusion obtained from the experiments applies to more challenging tasks. For example, does the conclusion also suitable for more challenging object categories, i.e. dogs, cats, fruits, etc., or more challenging tasks, i.e. tasks in 3D environments.

**Time Spent Reviewing:**

5 hours

---

> ### Author Response · Authors · 2021-08-10
> **Response to Reviewer cuD1**
>
> Thanks to Reviewer cuD1 for the helpful review!
>
> ## Idea too simple/findings are unsurprising/not a big contribution
>
> We agree that the contribution of the first half of our paper—the proposed setref/concept games—is quite straightforward in hindsight. However, we also believe that simple ideas can be non-obvious, impactful, and important.
>
> There is a long tradition in the emergent communication literature of carefully analyzing and manipulating environmental and architectural pressures to see what might give rise to more systematic communication in agents ([1](https://arxiv.org/abs/1906.02403), [2](https://arxiv.org/abs/1804.02341), [3](https://arxiv.org/abs/1804.03984), [4](https://openreview.net/forum?id=ZbXlSL_xwtA), [5](https://arxiv.org/abs/1705.11192), [6](https://arxiv.org/abs/2106.02067), etc). However, nearly all of this work operates within the framework of the classic Lewis referential game. Our paper makes the (simple in hindsight) claim that such games are inherently very limiting, and that trying to promote more general linguistic behavior in such a limited setting is “trying to fit a square peg into a round hole”, so to say. We hope our paper promotes a shift of perspective in the community on how best to promote and evaluate the emergence of more general linguistic phenomena in multi-agent systems.
>
> ## Tasks too easy; train on more diverse categories.
>
> First, we believe our choice of two visual domains is typical of, if not more complex than, similar work in the emergent communication literature, which often focuses exclusively on symbolic datasets or ShapeWorld/CLEVR-esque environments only ([1](https://arxiv.org/abs/1906.02403), [2](https://arxiv.org/abs/1804.02341), [3](https://arxiv.org/abs/1804.03984), [4](https://arxiv.org/abs/2004.09124), etc).
>
> One of reviewer cuD1’s specific concerns is that our agents are not trained on more difficult tasks involving a more diverse set of categories (e.g. dogs, cat, fruit), perhaps those in ImageNet. We stress that the Birds task is challenging for communication **precisely because the concepts are not diverse, and closely overlap.** This makes Birds ideal for our experiments.
>
> For example, imagine that the target concept is Laysan Albatross. The distractors are all birds, which can include 3 kinds of albatrosses, 8 varieties of seagulls, 7 kinds of terns, etc (all large white birds). Messages for the game must precisely describe the specific fine-grained visual features that distinguish Laysan albatrosses from Black-footed albatrosses or indeed any of the ~20+ large white bird species present in the dataset. In contrast, ImageNet categories are extremely diverse (orange, book, dog, lake). Diverse distractors make language easier to generate, and less interesting (compare just saying “orange” to “the white bird with the red tip on its beak and wide feathers”). Even sampling “close” ImageNet distractors would not help, since ImageNet contains fewer fine-grained visual categories, and even neighboring categories do not overlap that much in visual attributes (e.g. “orange”, “apple”, “banana”).
>
> This fine-grained setting is crucial to us because Birds gives us the underlying attributes for each bird, and there is rich overlap between attributes (e.g. birds that only differ by a specific body part). This gives us fine gradations of distance between concepts that we use to evaluate systematicity (e.g. topographic similarity, Table 1, Figure 3, etc). In contrast, ImageNet lacks annotated visual attributes, and any comparable dataset with a more diverse set of attributes would also mean a less overlapping set of attributes, which would make the analyses in our paper less effective.
>
> As final arguments for the difficulty of Birds, consider the corpus of language collected in a human image-captioning task for this dataset ([Reed et al., 2016](https://arxiv.org/abs/1605.05395)), which has over 5700 unique words and an average sentence length of 15.2 words. Finally, please see the experiments we ran in response to Reviewer Azs8, where we evaluate Resnet-18 (a very capable model) on the Birds task with both ground-truth language and a large (continuous) communication channel—these numbers demonstrate the difficulty of the task, even without emergent communication.
>
> ## Explore 3D (embodied?) environments.
>
> This is a great suggestion and an exciting avenue for future work, e.g. applying our insights to RL agents which must generalize across tasks, though one we believe is outside the scope of our paper.

---

> > ### Comment · Reviewer_cuD1 · 2021-08-13
> > **Thank you for the rebuttal**
> >
> > The authors provided explanations of the choice of their tasks. Even though I still think the tasks and environments are simple, this paper has other contributions that make me tend to accept it. I will keep my original rating.

---

### Official Review · Reviewer_13s9 · 2021-07-20

**Rating:** 7
**Confidence:** 4

**Summary:**

The current manuscript proposes to study the emergence of language when agents communicate with each other about generalizations over a set of objects, as part of a referential game. Experiments with object sets yield improved and more interpretable language that evolves between the agents. Additionally, the work also explores finding approximate logical operators in the emergent language.

**Limitations And Societal Impact:**

The work briefly mentions limitations (in conclusion) and broader impact.

**Main Review:**

**Strengths**
(S1) The manuscript does a good job of clearly stating the hypothesis about analyzing emergent language between agents in referential games about sets of objects. The paper is well-written and easy to understand. I thoroughly enjoyed reading the paper.

(S2) Experiments are well-rounded and back up claims introduced earlier in the paper. Further, analysis of emergent language through sunburst visualizations is insightful to understand the distribution of tokens. Though ambitious, Sec 7.1 to approximately search for operations in the emergence language is an interesting read. Kudos to the authors for attempting it.

**Weaknesses**
(W1) The paper seems to be missing a related work section. I urge the authors to include discussions around prior work related to other environmental pressures, placing this work in the context.

**Comments**
(C1) Are there any linguistic connections as to why generalizing to a set of objects results in more systematic and interpretable (as defined by metrics in the paper) emergent language?

(C2) How does the behavior of the agents and the emergent language change if the number of attributes and attribute values increases? The first dataset has multiple valued attributes but limited (5 shapes x 6 colors), whereas the second dataset has boolean attributes. Any insights on what happens if there are N attributes where each of them take M values, with increasing N and M?


**Time Spent Reviewing:**

2

---

> ### Author Response · Authors · 2021-08-10
> **Response to Reviewer 13s9**
>
> Thanks to Reviewer 13s9 for the detailed review!
>
> ## Related work section
>
> Thanks for pointing this out; see general response and our responses to related work inquiries from reviewers Azs8 and uEmD. We cited [Lazaridou et al. 2020](https://arxiv.org/abs/2006.02419) as a review of the literature that explores what kinds of environmental and/or architectural factors encourage the development of more compositional and systematic communication, but we agree that we should spend more time detailing specific works outside of that in lines 1-20. In contrast to much of this existing work, we tackle a separate aspect of the problem, which is the game objective itself.
>
> ## Connections between linguistics/generalizations and systematicity
>
> We referenced some ideas in the introduction in lines 23-30, but will elaborate more in the paper revision. One simple intuition is that humans learn categorical and conceptual mental representations that allow them to succeed in unseen environments: e.g. upon encountering a poisonous plant, we would like to avoid future poisonous plants, even if they do not look exactly like the one we encountered.
>
> This existing conceptual structure manifests in language: with the addition of a least effort minimization principle, human language is also able to convey rich generalizations over ideas and kinds in deceptively simple terms ([Tessler and Goodman, 2019](https://pubmed.ncbi.nlm.nih.gov/30762385/)), and generic language is a cornerstone of child language acquisition ([Gelman, 2004])(https://psycnet.apa.org/record/2004-12698-014). The human preference for generics naturally lends itself to simpler language: instead of describing particulars that apply only to a single object (e.g. the specific appearance of a  single poisonous plant, including its number of leaves, orientation, etc), we learn to discard extraneous information and convey only the relevant core attributes needed for category learning (e.g. bright blue berries). This simplicity would naturally manifest itself in terms of less diversity in generating utterances, alignment with core visual features, etc., that we measure in the paper.
>
> ## How does communication change as # of attributes/values vary?
>
> This is a good question. We ran some initial experiments investigating this question by adding additional shapes (cross, pentagon, hexagon, semicircle) and colors (cyan, magenta, purple, orange) to our ShapeWorld dataset, such that we can vary the (number of shapes, number of colors) in the dataset among the following values: (3, 4), (5, 6), (7, 8), (9, 10). We then ran agents with a vocab size of 22, i.e, sufficient to convey 9 shapes + 10 colors + 3 operators, on these dataset variants. Due to time constraints we were only able to run a single run of each of the following configurations, but the results are as follows (* indicates original setting in the paper)
>
> | Game Type | # shapes | # colors | Acc (Seen) | Acc (Unseen) | H(M \| C) | AMI(M, C) |
> |-----------|----------|----------|------------|--------------|-----------|-----------|
> | Ref       | 3        | 4        | 97         | 98           | 4.4       | 0.00      |
> | Ref       | 5*        | 6*        | 97         | 98           | 7.3       | 0.04      |
> | Ref       | 7        | 8        | 97         | 97           | 5.8       | 0.01      |
> | Ref       | 9        | 10       | 97         | 98           | 5.9       | 0.00      |
> | Setref    | 3        | 4        | 99         | 97           | 3.7       | 0.64      |
> | Setref    | 5*        | 6*        | 92         | 87           | 3.9       | 0.59      |
> | Setref    | 7        | 8        | 92         | 90           | 4.1       | 0.44      |
> | Setref    | 9        | 10       | 87         | 82           | 4.7       | 0.37      |
> | Concept   | 3        | 4        | 99         | 92           | 3.7       | 0.73      |
> | Concept   | 5*        | 6*        | 88         | 75           | 2.4       | 0.66      |
> | Concept   | 7        | 8        | 80         | 78           | 2.6       | 0.49      |
> | Concept   | 9        | 10       | 80         | 74           | 2.2       | 0.30      |
>
> We make the following preliminary observations:
>
> - As the number of attributes increases, communication gets harder (accuracy is lower across the board);
> - As the number of attributes increases, the systematicity of the communication protocol decreases somewhat, at least according to the information theoretic quantities H(M | C) and AMI(M, C). Part of this change is due natural test error in agents as the number of possible concepts expands combinatorially (from 128 concepts in the 3, 4 setting to ~1000 concepts in the 9, 10 setting). For example, it is possible that agents need to shift their communication more towards communicating a *continuous* value of color, rather than a discrete one, as the  number of colors increases, and colors get closer together in hue space. One exception to this trend is concept game entropy does not increase.
> - Regardless, across the board, setref and concept agents are more systematic than ref agents in every configuration of # shapes/# colors. Ref agents do not evolve more systematicity in their communication as the number of attributes and values increases.
>
> There is still more to investigate here; we will provide a more detailed analysis in the appendix of our paper revision.

---

> > ### Comment · Reviewer_13s9 · 2021-08-24
> > **Response to a author rebuttal**
> >
> > Thanks to the authors for the high quality rebuttal. They address all my concerns. I do believe that the current work has good contributions to the field of language emergence. I'd like to keep my rating of "Accept".

---

### Author Response · Authors · 2021-08-10
**General Response**

Thanks to all reviewers for the detailed and constructive feedback! We appreciate that all 4 reviewers are positive on the paper, and vote for acceptance or weak acceptance. Reviewers find the paper “didactic” (uEmD), easy to understand (13s9, cuD1), with well-rounded (13s9), sound, (uEmD), and good (cuD1) experiments, “insightful” (13s9) and novel (uEmD) visualizations, and overall “a number of interesting contributions” (Azs8). We hope our work provides insight into how agents might be encouraged to learn more abstract, generalizable visual concepts, and thereby more systematic and human-like languages.

We especially appreciate the positive feedback on the more exploratory Section 7.1 (ACRe), which reviewers found interesting (13s9, cuD1, Azs8), “ambitious” (13s9), and a helpful contribution to the community (Azs8). We hope ACRe will serve as an initial framework for more precisely investigating compositionality in emergent communication protocols.

Reviewers bring up several good questions and clarification points. **We summarize the reviews as follows**, as well as the actions we will take to improve the paper. **More detail is found in individual responses to each review.**

## Reviewers 13s9, Azs8, uEmD would like more discussion of related work.

Thanks to reviewers for pointing out connections from outside the emergent communication literature that could be discussed more, which as reviewer uEmD says would better constitute a “core contribution” lacking in the community. We will include a more detailed related work section with the additional space if accepted, including:

- Additional context on compositionality in emergent communication (13s9) elaborating on lines 15-20.
- Expanding on the cognitive motivations for communication of generalizations leading to more systematicity (13s9), elaborating on lines 24-30.
- Connections to prototypical/multi-instance/set representation learning (Azs8, uEmD).
- Connections to language automata (uEmD).

More detail in separate replies to each reviewer.

## Reviewer cuD1 has concerns about the simplicity of the contribution and the tasks.

- We agree that one of our contributions (the setref/concept games) is simple in hindsight, but believe that simple ideas can be meaningful and important.
- We also believe our tasks are typical for the emergent communication literature, are actually quite challenging due to the realistic visual input and overlap in concepts (for Birds), and are an appropriate choice for our setting because we need richly annotated dataset of visual attributes for analysis (cf. ImageNet). More details in the direct response.

## Reviewer Azs8 has concerns about the lower performance on setref/concept games, which may suggest that systematicity is not desirable in emergent communication.

Our detailed direct response to reviewer Azs8 can be summarized as follows:

- First, even if there is no benefit to generalization, there are many reasons to desire systematicity in emergent communication (chiefly, interpretability), and much of the literature similarly assumes systematicity to be an *intrinsic good*;
- Second, it is incorrect to interpret setref/concept game agents as having worse performance or less generalization ability than to reference game agents, *since the tasks are fundamentally different* (we realize we were unclear in our writing on this).
- **Ref agents do not learn any sort of human-like generalization at all!** They achieve high accuracy because the task permits them to learn trivial strategies based on low-level visual features, and not human-like abstractions. It is setref and concept agents that exhibit generalization behavior, as Table 2 shows.
- **To calibrate performance of our agents on setref and concept tasks**, we run experiments showing (1) agent performance given ground-truth (human) language and (2) agent performance with a continuous communication channel. These results show that our agents do very well with the emergent languages, often approaching the generalization limits of standard neural architectures on the task itself. In other words, imperfect performance of our agents is not solely due to inability to learn a sufficient language, but also due to reasonable limitations of our models on the tasks.

## Reviewers 13s9 and uEmD suggest small follow-up experiments/additional metrics. Others suggest minor presentational improvements.

We have conducted these experiments, have included results in the direct responses, and will include results in the paper revision. We will additionally incorporate all suggested improvements.

---

### Author Response · Authors · 2021-09-01
**Thanks!**

We'd like to thank all reviewers for providing detailed and constructive reviews, and additionally for engaging with our responses during the rebuttal phase. Each reviewer has provided concrete suggestions that will improve the quality of our paper!

---

### Decision · Program_Chairs · 2021-09-27

**Decision:**

Accept (Poster)

**Comment:**

This work points out limitations of existing reference games, and proposes using communication for generalization over (concept-level) target sets instead of single targets. Different types of targets, distractors and target sets are explored. The reviewers agree the current work presents several good and useful contributions to the field of emergent language. The authors did an excellent job in addressing any remaining concerns in the rebuttal.